# Generative learning facilitated discovery of high-entropy ceramic dielectrics for capacitive energy storage

Wei Li[1], Zhong-Hui Shen [1,2] ✉, Run-Lin Liu[2], Xiao-Xiao Chen[2], Meng-Fan Guo[3], Jin-Ming Guo[4], Hua Hao [1], Yang Shen [5], Han-Xing Liu [2], Long-Qing Chen [5] & Ce-Wen Nan [3] ✉

Dielectric capacitors offer great potential for advanced electronics due to their high power densities, but their energy density still needs to be further improved. High-entropy strategy has emerged as an effective method for improving energy storage performance, however, discovering new high-entropy systems within a high-dimensional composition space is a daunting challenge for traditional trial-and-error experiments. Here, based on phase-field simulations and limited experimental data, we propose a generative learning approach to accelerate the discovery of high-entropy dielectrics in a practically infinite exploration space of over $10^{11}$ combinations. By encoding-decoding latent space regularities to facilitate data sampling and forward inference, we employ inverse design to screen out the most promising combinations via a ranking strategy. Through only 5 sets of targeted experiments, we successfully obtain a $Bi(Mg_{0.5}Ti_{0.5})O_3$-based high-entropy dielectric film with a significantly improved energy density of 156 J cm$^{-3}$ at an electric field of 5104 kV cm$^{-1}$, surpassing the pristine film by more than eight-fold. This work introduces an effective and innovative avenue for designing high-entropy dielectrics with drastically reduced experimental cycles, which could be also extended to expedite the design of other multicomponent material systems with desired properties.

Dielectric capacitors capable of storing and releasing charges by electric polar dipoles are the essential elements in modern electronic and electrical applications such as hybrid electric vehicles, portable electronic devices as well as power pulse systems, owing to much higher power density than the electrochemical counterparts[1,2]. However, both ceramics possessing high dielectric constant and polymers featured by high breakdown strength face the dilemma that the energy density $U_e$ is much lower than that of chemical energy storage devices such as batteries[3,4]. Meanwhile, the lower energy density $U_e$ of dielectric materials greatly limits their applications and developments towards miniaturization and integration in the new era of the Internet of Things[5]. Therefore, it is of great significance to develop dielectric capacitors with higher energy density.

[1]State Key Laboratory of Advanced Technology for Materials Synthesis and Processing, Center of Smart Materials and Devices, Wuhan University of Technology, Wuhan 430070, China. [2]School of Materials and Microelectronics, Wuhan University of Technology, Wuhan 430070, China. [3]School of Materials Science and Engineering, State Key Lab of New Ceramics and Fine Processing, Tsinghua University, Beijing 100084, China. [4]Electron Microscopy Center, Ministry of Education Key Laboratory of Green Preparation and Application for Functional Materials, School of Materials Science and Engineering, Hubei University, Wuhan 430062, China. [5]Department of Materials Science and Engineering, The Pennsylvania State University, University Park, PA 16802, USA. ✉e-mail: zhshen@whut.edu.cn; cwnan@mail.tsinghua.edu.cn

For dielectric capacitors, the expression for the energy density is $U_e = \int_{P_r}^{P_m} E dP$. The simultaneous pursuit of a large maximum polarization $P_m$, a small residual polarization $P_r$ and a high breakdown strength $E_b$ is the key to realizing a high $U_e$[6]. In general, some ferroelectrics with large spontaneous polarization $P_s$, such as $BaTiO_3$ (BTO), $BiFeO_3$ (BFO), $PbZr_{1-x}Ti_xO_3$ (PZT), have shown great potential for achieving high $P_m$ (>50 $\mu$C cm$^{-2}$)[7,8]. However, a large number of polar dipoles in those ferroelectrics bring much stronger hysteresis effect, resulting in extremely large $P_r$ and bad voltage resistance (low $E_b$)[9]. Recently, high-entropy strategy[10,11] with local disorder have been proposed to enhance energy storage performance by modulating the mutual constraints among $P_m$, $P_r$, and $E_b$[10,12]. Here, the atomic configuration entropy $S_{config}$ is defined as $S_{config} = -R[(\sum_{i=1}^{N} x_i \ln x_i)_{cation-site} + (\sum_{j=1}^{M} x_j \ln x_j)_{anion-site}]$ with $R$, $N$ ($M$), and $x_i$ ($x_j$) representing the ideal gas constant, atomic species, and contents at the cation/anion sites, respectively[12]. As for high-entropy dielectrics (HEDs) with $S_{config} \geq 1.5\,R$, a local diverse polarization configuration inspired by ions with various valence states, ionic radii, and electronegativities can be achieved, resulting in smaller polar nanoregions (PNRs) with weak coupling and fast polarization response to an applied electric field thereby reducing $P_r$[2,13]. On the other hand, an increased lattice distortion can increase the crystalline energy to a special state that cannot be compensated by shrinking grain surface areas and inhibiting grain coarsening, which leads to the formation of fine grains or amorphous phases with enhanced $E_b$[5,14]. For example, ref. 12. reported a $Bi_2Ti_2O_7$-based high-entropy dielectric film with lattice distorted nano-crystalline grains and a disordered amorphous-like phase. As a result, the energy storage performance is substantially improved by the synergistic contributions of the enhanced breakdown strength and reduced polarization switching hysteresis. Chen et al. [14]. introduced local polymorphic distortions into (K, Na)NbO$_3$-based high-entropy ceramics to improve $E_b$ and delay polarization saturation, thereby achieving both large $U_e$ and high efficiency. Thus, high-entropy design is considered as an emerging and efficient method to improve the energy storage performance by optimizing the balance among $P_m$, $P_r$ and $E_b$.

However, it is noteworthy that HEDs comprising multiple chemical elements present an extensive compositional space. Thus far, the discovery of new HEDs with superior energy storage capabilities in traditional experiments has predominantly relied on empirical or trial-and-error methods, which leads to protracted and labor-intensive development cycles[14,15]. Consequently, efficiently and accurately identifying desired HEDs within high-dimensional composition spaces poses a formidable challenge. In recent years, the emergence of the fourth data-driven paradigm based on machine learning has thrived in accelerating the discovery and design of new materials[16,17]. By extracting and learning physical or chemical descriptors based on material characteristics, machine learning could predict and discover new knowledge and patterns to guide materials research[16,18]. But several challenges still persist. One of the main contradictions is the huge materials exploration space and the limited available data. When data is scarce, machine learning models would encounter problems such as poor generalization ability, overfitting, and high bias[19–21]. Generative learning, different from discriminative model, offers a method of generating new data by learning the underlying patterns or distributions from a given set of data. Thus, the power and significance of generative learning lies in its ability to create new data with similar characteristics, which has great potential in reinforcing limited materials data in an unsupervised or partially supervised way[17,21]. Therefore, drawing on this idea, we introduce generative learning into the process of machine learning-driven materials development to solve the major challenge of insufficient data volume by generating new dataset.

In this work, we construct a generative learning-based framework based on small experimental data to accelerate the discovery of HEDs with high energy density. To figure out the effect of configurational entropy on polarization response, we perform phase field simulations

to calculate polarization-electric field (P-E) loops and corresponding energy density of HEDs with different entropy values[22–24]. As an experimental example, we choose $Bi(Mg_{0.5}Ti_{0.5})O_3$ (BMT) because of its strong ferroelectric features and relatively good stability as pristine matrix to design HEDs by simultaneous multi-element doping of its A-site and B-site[25–27]. Taking 77 sets of experimental results as initial data, we build a generative learning model based on an encoding-decoding architecture with data reconstruction and artificial neural network (ANN) to find the potentially optimal high-entropy combinations[28,29]. The existing small sample data is then augmented with probabilistic sampling, where the elemental content of the A- and B-positions are retained to two decimal places and each position is summed up to equal 1. Thus, a possible space of nearly $10^{11}$ combinations is constructed to search for optimal combinations that satisfy the high entropy criterion. Then, we screen out top five compositions of prediction results among more than 2000 candidates, and five groups of targeted experiments are conducted to verify their potential in energy storage performance. Finally, a greatly improved $U_e$ of 156 J cm$^{-3}$ at 5104 kV cm$^{-1}$ has been obtained in $Bi_{0.87}La_{0.08}Sr_{0.05}Ti_{0.41}Mg_{0.39}Mn_{0.15}Zr_{0.05}O_3$ dielectric film, which is about eight times as much as that of pristine BMT (~18 J cm$^{-3}$). Our generative learning paradigm to design high-entropy dielectrics can also be extended to the design of other high-entropy functional materials with limited available data.

## Results
### Phase-field simulations of high-entropy effect
To theoretically evaluate the high-entropy engineering on improving the energy storage performance of dielectrics, we first perform phase-field simulations to study the local polarization response and macroscopic P-E loops with different $S_{config}$[1]. At the atom and lattice levels, $S_{config}$ increases with the entry of foreign atoms into the equivalent position, leading to higher atomic disorder and lattice distortions or oxygen-octahedral distortions due to the differences in atomic size, mass, and electronegativity[30], as illustrated in Fig. 1a. As a result, random local strains or electric fields appear at nanoscale, which may disrupt microdomains into polar nanoregions. The larger the $S_{config}$, the more disordered the polar dipole distribution. On the above basis, we build different phase-field models with different degrees of dipole disorder in $BiFeO_3$-based dielectrics, as described in *Methods*. As shown in Fig. 1b, as $S_{config}$ increases from 0.69 R to 1.64 R, the polar regions are broken into increasingly disordered and decreasing areas. At low entropy, the size of polar regions is large and the dipole orientation distribution is uniform and orderly. As $S_{config}$ increases to 1.64 R, the polar regions are broken into much smaller areas and more disordered dipoles appear with a very chaotic state (Fig. 1b). Therefore, for high-entropy dielectrics with strong local heterogeneity, the dipole switching would become much easier and thus the ferroelectric hysteresis effect would be greatly alleviated[5]. This microscopic configuration change has also been verified by our simulation results of P-E loops in Fig. 1c. As $S_{config}$ increases, the P-E loop curves gradually shift from ferroelectric to relaxor-like behavior. As the shaded area displayed in Fig. 1c, $U_e$ raises substantially with the increase of $S_{config}$. The improvement of $U_e$ caused by entropy increase has also been confirmed in other systems of $BaTiO_3$ and $PbTiO_3$-based dielectrics. As shown in Fig. 1d, $U_e$ of all three systems show a similar increase trend with $S_{config}$ changing from 0.69 R to 1.64 R. The improvement of energy storage performance by high entropy design concluded by our simulations results is also consistent with the existing experiments[12,24,31]. Therefore, high-entropy engineering, as a universal design strategy, has shown great potential in enhancing the energy storage performance by modulating local dipole configuration[32].

### Machine learning-driven high-entropy design
To experimentally realize the high-entropy design in dielectrics, based on our knowledge and experience in the field of materials, we

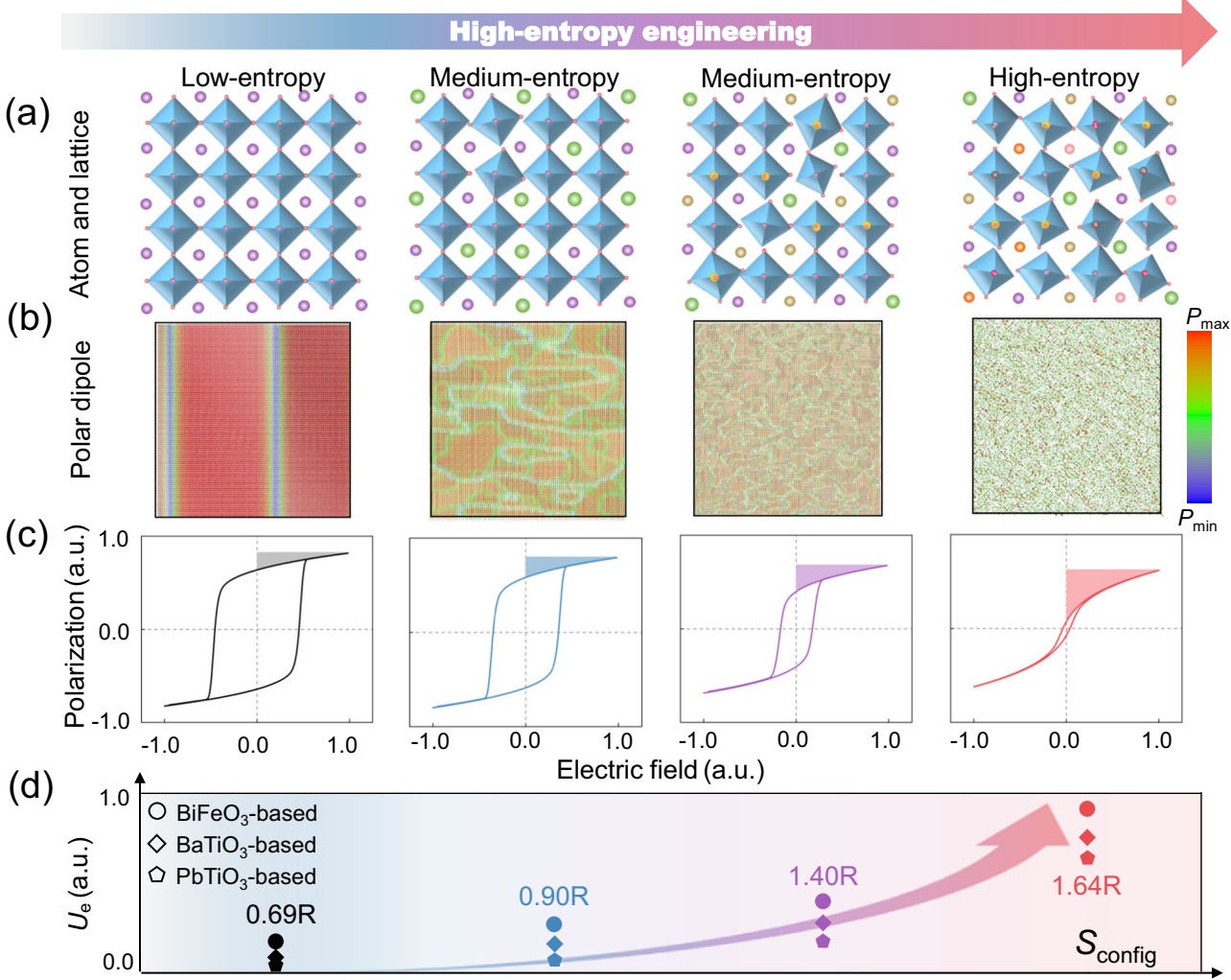

**Fig. 1 | Phase-field simulations of configuration entropy (Sconfig) effect on energy storage performance. a** Schematics of atomic disorder and lattice change in perovskite structures from low entropy to high entropy. **b** Corresponding polarization distributions in BiFeO$_3$-based dielectrics with different $S_{config}$. **c** Normalized *P-E* loops under the same applied electric field, where the shaded areas represent the dischargeable energy density $U_e$. **d** Normalized $U_e$ of BFO, BTO, and PTO-based dielectrics with four $S_{config}$ of 0.69 R, 0.90 R, 1.40 R, and 1.64 R.

carry out an expert-orientated trial-and-error method to find high-entropy systems with suitable elements and their contents. Taking Bi(Mg$_{0.5}$Ti$_{0.5}$)O$_3$ as initial matrix, we have prepared 77 systems of Bi$_{(1-a-b-c)}$La$_a$Sr$_b$Ca$_c$(Mg$_{0.5}$Ti$_{0.5}$)$_{1-d-e-f}$Mn$_d$Zr$_e$Hf$_f$O$_3$, including 48 sets of high-entropy combinations by introducing different elements in the A-site and B-site, as detailed at Supplementary Table 2. Unfortunately, the time-consuming and labor-intensive repetitive exploration experiments did not help us find a high-entropy system with excellent energy density. The highest $U_e$ among them is 87 J cm$^{-3}$, just over three times as high as pristine BMT. This is because the huge exploration space of potential high-entropy systems makes it difficult to find the optimal composition quickly and accurately, which is also a common challenge faced by current high-entropy materials design[33]. In order to efficiently implement high-entropy design, we propose a data-driven pattern with machine learning screening and directed experiment validation to accelerate the discovery process of high-entropy dielectrics with high energy storage performance. As displayed in Fig. 2a, the machine learning framework consists of three parts: (i) the generative model with generation of the latent space z, (ii) classification and sampling of compositions, and the predictive model of (iii) forward inference and inverse design. First, before embarking on a machine learning-driven high-entropy design, having enough training data is the most basic requirement. Obviously, the

existing experimental data is not enough to support the model to show good global generalization ability in such a huge exploration space. Thus, we develop a generative model (GM)-based framework with a neural network for the encoder-decoder of HEDs[34]. Here, we construct encoders with the types and contents of dopant elements in the A-site and B-site Bi$_{(1-a-b-c)}$La$_a$Sr$_b$Ca$_c$(Mg$_{0.5}$Ti$_{0.5}$)$_{1-d-e-f}$Mn$_d$Zr$_e$Hf$_f$O$_3$ (C-n, n = 1,2,3…), where the main dopant elements in the A-site are La, Sr and Ca, and the main dopant elements in the B-site are Mn, Zr and Hf, and the content varies from 0 to 0.01 in each case. Using the elemental composition information of every initial film as the input data, a latent feature space z that can be used to potentially represent the dielectric information variables is generated, and then decoding z can regenerate the reconstructed compositions. Supplementary Fig. 1, Supplementary Tables 3 and 4 show the reconstructed results of the compositions. The model is also analyzed for its ability to extract high-entropy compositions represented as low-dimensional latent variables, as shown in Supplementary Fig. 1, where the smooth curve indicates that the model eventually stabilizes. Thus, we could get a physically meaningful and informative z with a large amount of information about the features of high-entropy dielectrics. Then, based on the screening of targeted metrics, we build a classifier that can recognize high and low energy density (Supplementary Fig. 2) to construct a composition-$U_e$ relationship from a small dataset.

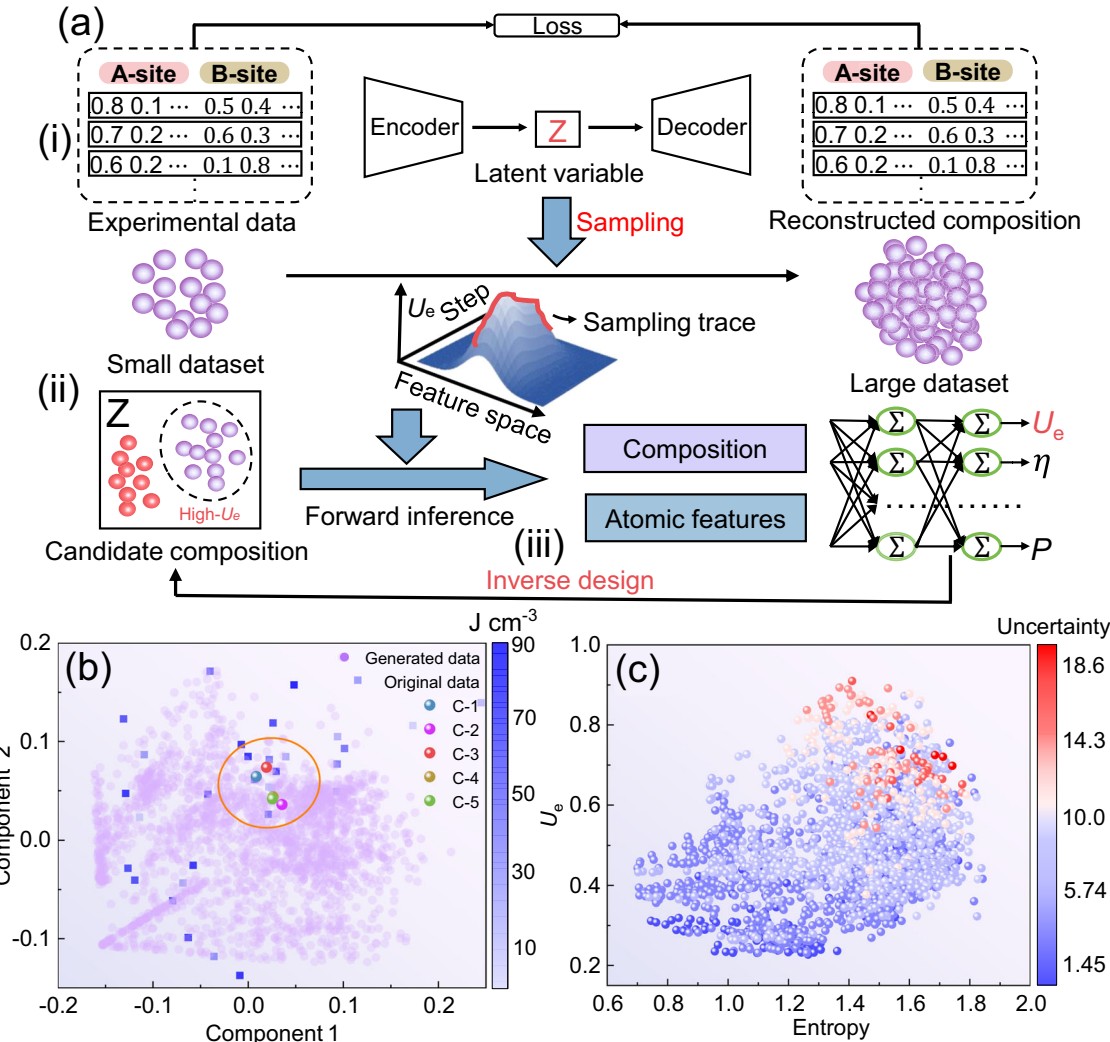

**Fig. 2 | Overview of generative learning framework for high-entropy design. a** A generative learning model for the design and discovery of high-entropy dielectric materials. The framework is divided into three steps: (i) generation of the latent space $z$ (ii) classification and sampling of compositions, and (iii) forward inference and inverse design. **b** Latent space distribution of the different components, purple circles represent the generated 2144 sets of high-performance data, blue squares represent the original experimental data, and solid spheres of different colors represent the five new sets of components predicted by the model. **c** Entropy versus normalized $U_e$ of candidate materials, where the color of the data points represents their uncertainty.

Because the space of C-n compositions is too large with about $10^{11}$ possible combinations, we analyze known high-performance compositions in the low-dimensional space by Gaussian mixture model (GMM) (Supplementary Fig. 3) and Markov Chain-Monte Carlo Sampling (MCMC) to inversely generate unseen other compositions with similar high performance[35,36]. Next, based on the inverse target design, we use a regression model to further inference the $U_e$ of the candidate compositions generated by GM. With this regression model integrating the advantages of Artificial Neural Networks (ANN) and Light Gradient Boosting Machine (Light GBM) methods, the composition-descriptor-performance relationships can be established to achieve fast and large-scale combinatorial inference[37–40]. The physical descriptors in this model can be seen in Supplementary Table 5. The scarcity of experimental data, as well as the difficulty of generalizing the constructed model globally and the highly non-linear nature of the composition-performance relationship would lead to instability in the regression model[41]. Thus, to improve the efficiency of data mining as well as the robustness of the regression model, we define a mining capacity coefficient λ that ranks the sum of the predicted values and their uncertainty weights, similar to a Bayesian optimization, to guide the discovery of desirable compositions. In

traditional active learning strategies, the uncertainty tends to favor the space of compositional combinations whose predicted values have higher variance. We have improved uncertainty trade-off with the goal of predicting dense, stable compositional data in one go. We rank combinations by λ, as this strategy magnifies the gap between different candidates[42,43]. The ranking-based strategy ensures that candidate portfolio selection is less affected by model inaccuracies and provides a systematic way to combine model predictions and uncertainty.

Based on 77 sets of BMT-based experiment results as initial data, we generate 2144 sets of high-performance systems with energy densities greater than $65\,J\,cm^{-3}$, and then select the top five sets for targeted experiments. The GM-generated potential space $z$ is visualized by principal component analysis (PCA) downscaling, as shown in Fig. 2b, where the blue squares denote the original experimental data, corresponding to the color bar on the right side. Purple circles denote the 2144 candidate sets of high-energy-density potential data sampled from the classifier ($U_e > 65\,J\,cm^{-3}$). Solid spheres with different colors indicate the new five components generated by model predictions (C-n, $n = 1,2,3,4,5$). Their elemental species and contents, entropy values, and uncertainties are shown in Supplementary Table 6. The five new

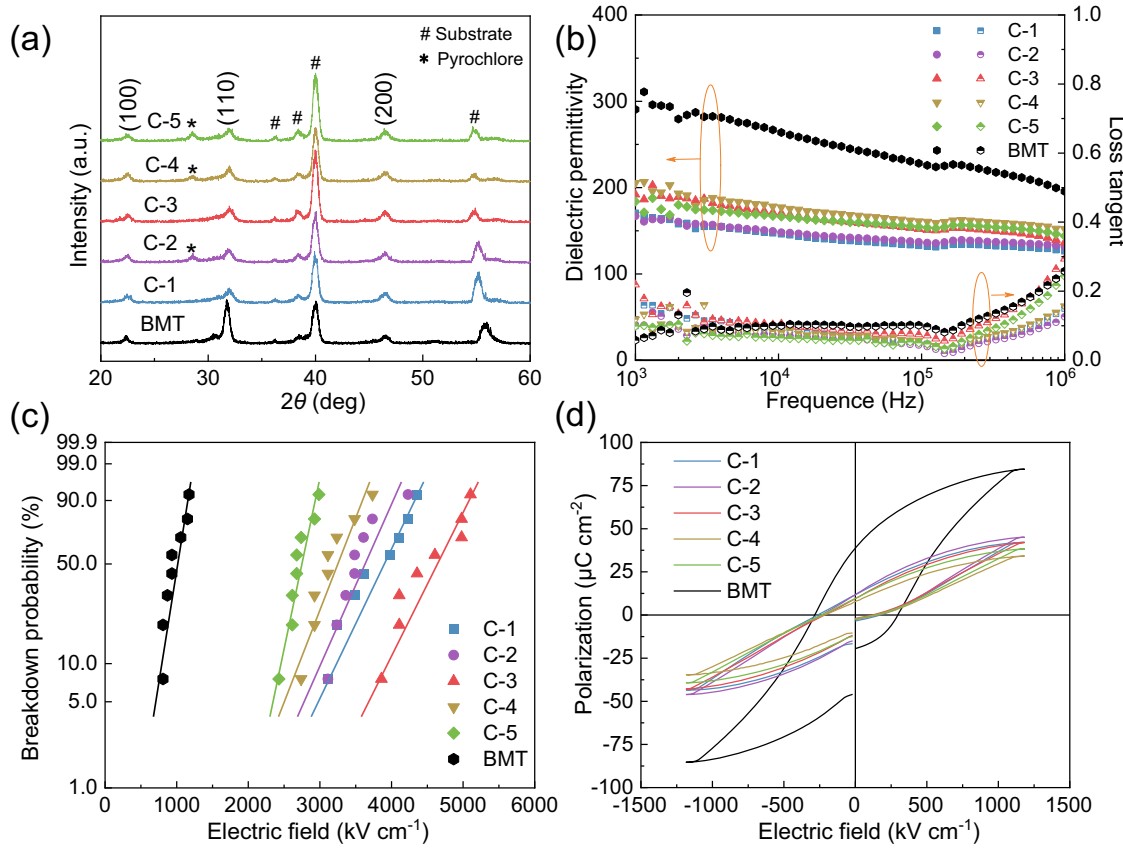

**Fig. 3 | Analysis of phase structure and electrical properties. a** GI-XRD patterns of BMT and C-n films. **b** Frequency-dependent dielectric properties from 1 kHz to 1 MHz. **c** Weibull distribution analysis of the breakdown strength, where corresponding Weibull modulus $\beta$ are 7.76, 10.19, 10.2, 11.8, 10.51 and 16.97, respectively. **d** P-E loops at an electric field of 1200 kV cm$^{-1}$ and a frequency of 1 kHz.

compositions are located in the middle dense region (inside the yellow circle), indicating that the energy density of the compositions in this region is more in line with our expectations. The relationship between the entropy value and the normalized $U_e$ of each composition in the candidate space predicted by the regression model is shown in Fig. 2c. As the entropy value increases, the combination of elements becomes more and more complex, and therefore the uncertainty increases, which the overall $U_e$ of the screened composites tends to increase. This is highly consistent with our previous phase-field simulation results. Based on the screening results by the ranking strategy, we carry out directed experiments to verify the theoretical predictions. More details about experiment results will be introduced in the following section. Encouragingly, the top five high-entropy designs we screened all gain in energy storage performance, especially C-3 film with the highest $U_e$ of 156 J cm$^{-3}$ and $E_b$ of 5104 kV cm$^{-1}$, which is about eight times more than that of BMT. This coincides with the results of our phase-field simulations, demonstrating the feasibility and effectiveness of the high-entropy strategy on improving energy storage performance. Thus, the machine learning strategy in this work successfully accelerate the discovery of high-performance HEDs by only 5 sets of directed experiments.

### Directed experiments and electrical characterization

Guided by phase-field simulations and machine learning predictions, we conduct directed experiments for the five screened compositions of BMT-based high-entropy films by chemical solution deposition (CSD) with the thickness of about 160 nm (Supplementary Fig. 4). For each high-entropy film, different annealing temperatures were also explored to find the optimal energy storage performance. As shown in Supplementary Fig. 5, energy storage densities, breakdown strengths,

and maximum polarization values were displayed at different annealing temperatures. At the optimal annealing temperature of each component (C-n, $n = 1,2,3,4,5$), we further performed the relevant structural and morphological characterization as well as the electrical property testing. Figure 3a shows the grazing incidence x-ray diffraction (GI-XRD) pattern of the BMT-based films, where C-1 and C-3 formed pure perovskite phases consistent with BMT, and pyrochlore phases appeared in C-2, C-4 and C-5. Meanwhile, the positions of C-n films and BMT at the (100) and (110) peaks remain apparently unchanged, indicating that no lattice substitution has occurred even in the presence of elemental doping[27]. However, it can be clearly seen at (110) that the peak intensities of C-n films are significantly weaker than that of BMT, and the diffraction peaks gradually become broader. Those changes may be attributed to that the high entropy doping strategy leads to the grain refinement or the increase in the proportion of amorphous phases as a result of the deterioration of crystallinity[12,44]. This is also evident from the scanning electron microscope (SEM) of different films after annealing (Supplementary Fig. 6), which show denser microstructures with nanograins down to a few nanometers in Supplementary Fig. 7. As the high resolution transmission electron microscope (HR-TEM) results shown in Supplementary Fig. 8a, b, the lattice diffraction fringes can be observed in the area of the yellow circles, which are thus determined to be crystalline, and the rest amorphous. With the increase of entropy from BMT to C-3, the percentage of local amorphous phase increased from 10% to 45% (Supplementary Fig. 8c). More details about the experimental preparation and characterization are described in *Methods*.

To understand the electrical properties of five high-entropy BMT-based films, we first measured the dielectric constant and dielectric loss under different frequencies from 1 kHz to 1 MHz, as shown in

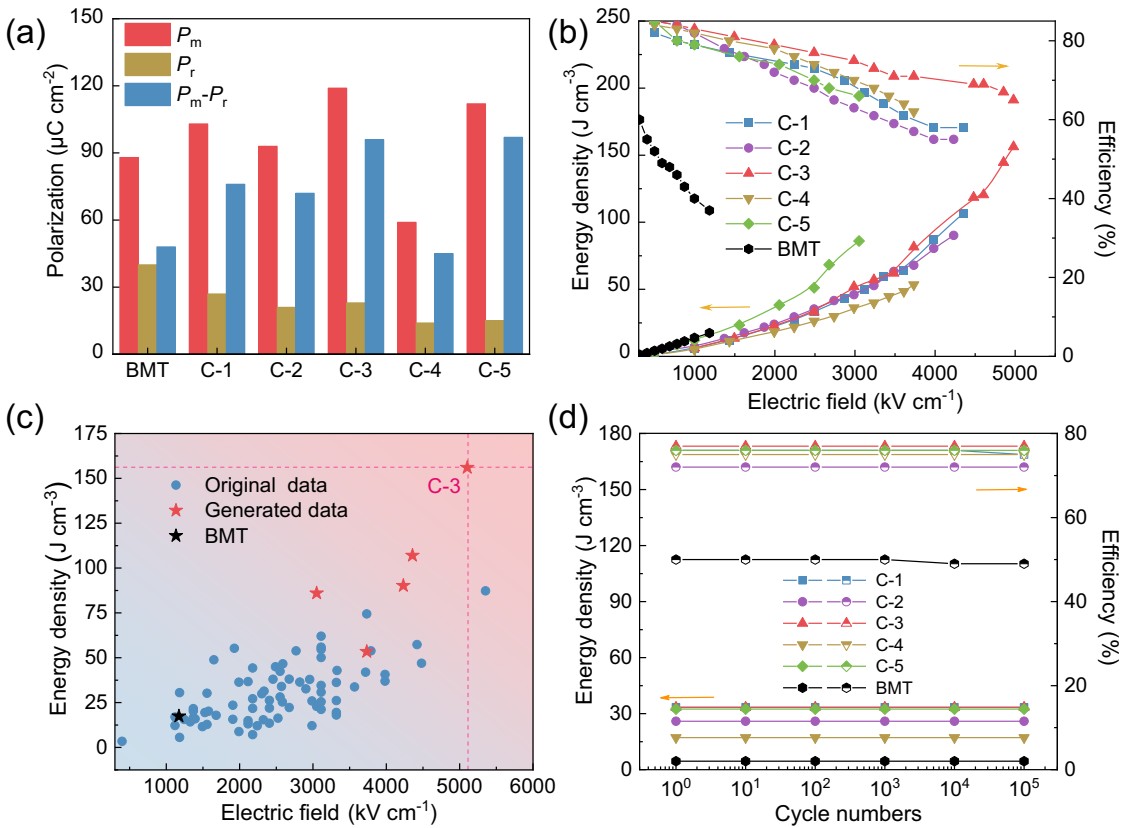

**Fig. 4 | Energy storage performance and cycling stability. a** $P_m$, $P_r$ and $P_m$-$P_r$ for each composition under their breakdown strength $E_b$. **b** Energy density and efficiency at the electric fields up to $E_b$. **c** Comparisons of energy densities and breakdown strengths between the original experimental compositions, the generated five experimental compositions, and BMT. **d** Charging-discharging reliability of different films at an electric field of 2000 kV cm$^{-1}$ and 1000 kV cm$^{-1}$ for BMT.

Fig. 3b. As a whole, the dielectric constants of C-n films are lower than that of BMT, decreasing from 310 of BMT to 190 of C-3 at 1 kHz, which is in line with the trend of polarization reduction observed in the phase-field simulations. This decrease may be due to the high entropy doping that refines the grains and introduces an excess of amorphous phases[45]. Over a certain frequency range of 1 kHz to 1 MHz, we observe higher dielectric constant stability, i.e., the variation of dielectric constant is ~ 10% for C-n films and 30% for BMT film. In addition, we find that the loss tangents of C-n films are somewhat suppressed in the measured frequency range. At 100 kHz, the dielectric loss has a relaxation peak with frequency, probably due to the grain refinement[27,45]. We ascribe the evolution of dielectric properties to the emergence of nanocrystals and the increase of amorphous phases, which are usually considered to have linear polarization characteristics with low dielectric constant, low loss, and good frequency stability[12,46]. As we discussed above, the maximum electric field that dielectrics can withstand is one of the important factors (denoted by $E_b$) to determine the maximum energy density, which is also a key performance index for high-entropy design to improve. To compare the voltage resistance characteristics, we analyze $E_b$ of BMT and C-n films using the Weibull distribution law, as shown in Fig. 3c. The Weibull modulus $\beta$ comes from the slope of the fitted lines, indicating the reliability of $E_b$ distribution and reproducibility of the films[47]. The Weibull modulus $\beta$ values of BMT and C-n (n = 1, 2, 3, 4, 5) films are 7.76 and 10.19, 10.2, 11.8, 10.51, and 16.97, and $E_b$ are 1173 and 4357, 4232, 5104, 3734, and 2987 kV cm$^{-1}$, respectively. To further explore the reason why high-entropy design enhances $E_b$, we provide the leakage current densities of BMT and C-n films under different applied electric fields[48]. As shown in Supplementary Fig. 9, it can be seen that the leakage current densities of C-n films at high electric fields are much lower than that of BMT film, which is nearly two orders of magnitude lower. For example,

at an electric field of 1000 kV cm$^{-1}$, the leakage current density of C-3 film is $4.01 \times 10^{-6}$ A cm$^{-2}$, while that of BMT is $3.97 \times 10^{-4}$ A cm$^{-2}$. Therefore, the high-entropy design of C-n films exhibit lower leakage current densities and higher $E_b$, which is beneficial to improve the maximal energy density. Then, we measured the $P$-$E$ loops of BMT and C-n films at 1500 kV cm$^{-1}$ to further evaluate their potential for energy storage performance, as shown in Fig. 3d. The results show that the low-entropy BMT film has a lower $\eta$ of 45% due to its higher $P_r$ (~40 μC cm$^{-2}$) and strong polarization switching hysteresis. While for those films with high-entropy design, all C-n films exhibit weaker hysteresis effect with enhanced relaxation-like properties, reduced residual polarization $P_r$ (~10 μC cm$^{-2}$), and much improved efficiency ($\eta$ ~ 75%). To understand the huge difference of polarization response between BMT and C-n films, we also characterized the local domain structures by piezo-response force microscopy (PFM) in two representative films of BMT and C-3 sample, as shown in Supplementary Figs. 10 and 11. Compared to the pure BMT, greatly reduced domain size and amplitude can be clearly identified in high-entropy C-3 film, which can be an important reason for the slim $P$-$E$ loops. Therefore, high-entropy design facilitates the advantages of improved insulation, reduced leakage current and improved breakdown strength, etc.

## Energy storage properties and cycling stability
In order to better evaluate the energy storage properties of C-n films, we plot the polarization information of each composition at their respective $E_b$, as shown in the bar chart of Fig. 4a. Pure BMT film has high $P_m$ (88 μC cm$^{-2}$) and also high $P_r$ (40 μC cm$^{-2}$), and the $P_m$-$P_r$ value is much lower than high-entropy films. This is because high-entropy design brings an increase in $P_m$ at higher $E_b$ but suppressed $P_r$, resulting in larger $P_m$-$P_r$ value. For example, the $P_m$ of C-3 film reaches 119 μC cm$^{-2}$ and $P_r$ is only 23 μC cm$^{-2}$, and the $P_m$-$P_r$ values of BMT and

C-n films are 48 and 76, 72, 96, 45, 97 μC cm$^{-2}$, respectively. In this work, the substitutions of Sr$^{2+}$ (1.44 nm) for Bi$^{3+}$ (1.36 nm) sites and Zr$^{4+}$ (0.72 nm) and Mn$^{2+}$ (0.67 nm) for Ti$^{4+}$ (0.605 nm) and Mg$^{2+}$ (0.72 nm) sites may distort the local lattices and thus forms local polarized nanoregions, which is beneficial to reduce the domain motion potential barrier, and significantly enhance the polarized region's mobility with increased $P_m$ and decreased $P_r$[49]. Consistent with our phase-field simulations, the high-entropy doping strategy plays a role in synergistically regulating $E_b$, $P_m$, and $P_r$, which is conducive to improving the energy storage performance of dielectrics for broader applications[50,51]. Based on their $P$-$E$ loops (Supplementary Fig. 12), we calculate the energy density $U_e$ and efficiency $\eta$ as shown in Fig. 4b. For pure BMT film, the maximal $U_e$ is only 18 J cm$^{-3}$. While for C-n films with high-entropy design, all maximal $U_e$ have been improved and the highest maximal $U_e$ reaches 156 J cm$^{-3}$ at 5104 kV cm$^{-1}$ for C-3 film, which is more than eight times higher than that of BMT. The high $U_e$ values may be attributed to the low loss (small $P_r$) and high entropy-induced lattice distortion, grain refinement, and amorphous phases[12,31,51]. Meanwhile, we compare the $U_e$ and $E_b$ of original experimental samples used for machine learning with BMT and C-n films ($n$ = 1, 2, 3, 4, 5), as shown in Fig. 4c. It can been seen that the $E_b$ of those high-entropy films by directed experiments have been greatly improved up to above 4000 kV cm$^{-1}$, leading to higher maximal $U_e$, as more pentagrams located in top right corner of Fig. 4c. Thus, it not only shows that the high-entropy strategy has great potential for improving energy storage performance, but also validates the feasibility of our approach of generative learning to efficiently find ideal high-entropy compositions with high energy densities.

Considering the practical applications, we tested the charging-discharging cycling reliability of the films under an electric field of 2000 kV cm$^{-1}$, as shown in Fig. 4d. The cycling reliability of pure BMT film is stable at 1000 kV cm$^{-1}$ but cannot withstand 2000 kV cm$^{-1}$. The energy storage performance of C-n films remained stable after $1 \times 10^5$ cycles and generally maintained good stability without obvious deterioration. For example, the C-3 film exhibits nice fatigue durability of at least $1 \times 10^5$ cycles with $U_e \sim$ 33.36 J cm$^{-3}$ and $\eta \sim$ 77% at 2000 kV cm$^{-1}$. We also evaluated performance stability over a wide temperature range of 20 to 150 °C, as shown in Supplementary Fig. 14a. Pure BMT film has poor stability with $U_e$ changing from 4.53 to 3.8 J cm$^{-3}$ and $\eta$ changing from 50% to 40% under an electric field of 1000 kV cm$^{-1}$, and is rapidly punctured after warming up under an electric field of 2000 kV cm$^{-1}$. The C-n films behave more stable under an electric field of 2000 kV cm$^{-1}$ and the $U_e$ of C-3 film changes from 33.36 to 29.36 J cm$^{-3}$ and $\eta$ from 77% to 70%. Therefore, high-entropy films have better temperature stability compared to the BMT film. In addition, we also evaluated their frequency stability, as shown in Supplementary Fig. 14b. It can be seen that at an electric field of 2000 kV cm$^{-1}$, all C-n films are stable and C-3 film keeps unchanged $U_e$ of ~4.53 J cm$^{-3}$ and $\eta$ of ~77%. As a result, all C-n films prepared in this work are characterized by low loss, high resistivity, and high breakdown strength, as well as excellent cycling reliability and temperature and frequency stability.

## Discussion

In summary, we achieve a theory-to-experiment methodological path for the rational design of high-entropy BMT-based dielectrics with high energy density. First, by performing phase-field simulations, we have predicted that the macroscopic energy storage performance could be adjusted by local dipole configuration in high-entropy dielectrics. To accelerate the discovery of ideal high-entropy combinations, a comprehensive machine learning framework has been developed with the generative learning and regression model to guide the directed experimental preparation of HEDs. Using limited experimental results, we use generative learning to search potential combinations through a large number of exploration space and then screen out five most promising high-entropy compositions by the ranking strategy. Finally, by

conducting only 5 sets of directed experiments, we obtain a new high-entropy system of Bi$_{0.87}$La$_{0.08}$Sr$_{0.05}$Ti$_{0.41}$Mg$_{0.39}$Mn$_{0.15}$Zr$_{0.05}$O$_3$ with a more than eight times higher $U_e$ (-156 J cm$^{-3}$) than pure BMT film. With both excellent fatigue properties and temperature and frequency stabilities, those high-entropy films also show great potential for wide use in energy storage capacitors. Based on the machine learning-driven patterns, we efficiently find the desired high-entropy composites with high energy storage performance using very sparse experimental data. This method also provides us with a significant reduction in overall experimental cycle time and opens up a new avenue for designing those material systems with complex components.

## Methods
### Phase-field simulation
The spontaneous polarization $P$ is selected to describe the temporal evolution of the polarization field and the domain structure to solve the three-dimensional time-dependent Ginzburg-Landau(TDGL) equation in the phase-field simulation:

$$\frac{\partial P_i(r,t)}{\partial t} = -L \frac{\delta F}{\delta P_i(r,t)} + \xi_i(r,t), (i = 1,2,3) \qquad (1)$$

where $L$ is the kinetic coefficient related with the domain wall migration rate, $F$ is the total energy, $r$ is the spatial position, $t$ is time, $P_i(r, t)$ is the polarization intensity at a certain space position and a certain time, $\xi_i(r,t)$ is the impact of thermal noise, which conforms to a random Gaussian distribution.

The total energy $F$ includes the Landau bulk free energy, gradient energy, elastic energy, and electrostatic energy, as follows:

$$F = \int_V [f_{bulk} + f_{grad} + f_{elas} + f_{elec}] dV \qquad (2)$$

where $V$ is the volume of the system, $f_{bulk}$, $f_{grad}$, $f_{elas}$, $f_{elec}$ represents the Landau bulk free-energy, gradient energy, elastic energy, and electrostatic energy, respectively.

In terms of polarization, the bulk free energy for a stress-free ferroelectric can be formulated as a six-order expansion, as follows:

$$\begin{aligned} f_{bulk} &= a_1(P_1^2 + P_2^2 + P_3^2) + a_{11}(P_1^4 + P_2^4 + P_3^4) + a_{12}(P_1^2 P_2^2 + P_1^2 P_3^2 + P_2^2 P_3^2) \\ &+ a_{112}[P_1^4(P_2^2 + P_3^2) + P_2^4(P_1^2 + P_3^2) + P_3^4(P_1^2 + P_2^2)] \\ &+ a_{111}(P_1^6 + P_2^6 + P_3^6) + a_{123}P_1^2 P_2^2 P_3^2 \end{aligned} \qquad (3)$$

where $a_1$, $a_{11}$, $a_{12}$, $a_{111}$, $a_{112}$, $a_{123}$ are all Landau energy coefficients, which associated with the thermodynamic behaviors of bulk phases.

Owing to the contribution of domain walls, the gradient energy $f_{grad}$ is expressed as follows:

$$\begin{aligned} f_{grad} &= \frac{1}{2} G_{11}(P_{1,1}^2 + P_{2,2}^2 + P_{3,3}^2) + G_{12}(P_{1,1}P_{2,2} + P_{2,2}P_{3,3} + P_{1,1}P_{3,3}) \\ &+ \frac{1}{2} G_{44}[(P_{1,2} + P_{2,1})^2 + (P_{2,3} + P_{3,2})^2 + (P_{1,3} + P_{3,1})^2] \\ &+ \frac{1}{2} G_{44}'[(P_{1,2} - P_{2,1})^2 + (P_{2,3} - P_{3,2})^2 + (P_{1,3} - P_{3,1})^2] \end{aligned} \qquad (4)$$

where $G_{ij}$ is gradient energy coefficient, $P_{i,j}$ is $\frac{\partial P_i}{\partial r_j}$.

The elastic energy $f_{elas}$ has the following expression:

$$f_{elas} = \frac{1}{2} c_{ijkl} e_{ij} e_{kl} = \frac{1}{2} c_{ijkl}(\varepsilon_{ij} - \varepsilon_{ij}^0)(\varepsilon_{kl} - \varepsilon_{kl}^0) \qquad (5)$$

where $c_{ijk}$ is the elastic stiffness tensor, $e_{ij}$ is the elastic strain, $\varepsilon_{ij}$ is the total strain and $\varepsilon_{ij}^0$ is electrostrictive stress-free strain. In high-entropy

ceramics, a local stochastic strain field is introduced to consider the doping effects of various chemical elements:

$$\varepsilon_{ij}^0 = Q_{ijkl}P_kP_l + cx \tag{6}$$

where $x$ represents the local strain coefficient. The concentration $c$ of doping which follows the Gaussian distribution in different entropy ceramics is listed in Supplementary Information.

For a given domain structure, the electrostatic energy $f_{elec}$ consist of applied external electric field and electric field which is formulated as follows:

$$f_{elec} = -P_iE_i^{ex} - \frac{1}{2}E_i^{in}P_i \tag{7}$$

where $E_i^{ex}$ is the applied external electric field, $E_i^{in}$ is the electric field induced by the dipole moment in the sample. The detailed parameters for each material are listed in supplementary Information.

## Machine learning

The encoder-decoder architecture was designed to represent the composition of the dielectric materials in an unsupervised manner. The main idea was to optimize a loss function(L), which is the weighted sum of maximum mean discrepancy between the $z$ and prior distribution and binary cross-entropy of input-output pair.

$$L = \|E_X \sim P[\varphi(X)] - E_Y \sim Q[\varphi(Y)]\|_H - \sum_{i=1}^n P[\varphi(X)]\log Q[\varphi(Y)] \tag{8}$$

where $P$ and $Q$ are two distributions of different dataset. The MMD is defined by a *feature map* $\varphi$: $S \rightarrow H$, $H$ is defined as the reproducing kernel Hilbert space. Such loss function is able to accurately represent the difference between the latent distribution and the prior distribution. We trained one encoder-decoder for A and B sites respectively, with the number of neurons per layer of the encoder being 80, 64, 48, and 2, while the decoder using exactly the opposite structure. Each layer is nonlinearly transformed using the layernorm and ReLU activation functions, with a batch size of 10, a learning rate of $1 \times 10^{-3}$, and an epoch of 300. The latent space of A and B sites are dimensionally reduced using PCA and visualized in Fig. 2b.

The GMM assumes the data is composed of multiple Gaussian distributions, which was used to model and estimate the distribution density of $z$. The optimal number of Gaussian clusters is usually determined by the empirical elbow method. ANN classifier was trained with two layers of simple neural network to identify raw samples with a high energy density. In order to make the classifier more precious, k-fold cross-validation was used. Then, we utilized Metropolis-Hasting MCMC in the latent space $z$ based on Markov chain, aiming to generate numerous data according to accept-reject sampling.

ANN, inspired by neural networks, excels at learning complex patterns, while LightGBM is a tree-based gradient boosting framework, efficient for handling high-dimensional features. This work integrates both to enhance predictive accuracy and robustness in regression tasks, among which the main hyperparameters were optimized by random searches followed by 10 rounds of Bayesian Optimization (BO). To eliminate the dimensional impact and enhance the stability of the regression model, all atomic properties were normalized. Moreover, predicted values and uncertainty were calculated by top fifty models from BO to reduce the incidental errors. We then measured the value of λ through a ranking strategy, as shown in Eq. 9, where α denotes predicted value, $\beta$ denotes uncertainty. The program was written using Pytorch and Sklearn. BO is used in the bayes_opt library in python.

$$\lambda = \alpha * rank(U_{Predict}) + \beta * rank(U_{Uncertainty}) \tag{9}$$

## Sample preparation

The high-entropy dielectric films of C-n ($n = 1,2,3,4,5$) were fabricated by chemical solution deposition method. Bismuth nitrate pentahydrate (Aladdin, 99%; with 10% excess to compensate the volatilization of Bi during heat treatment), lanthanum acetate sesquihydrate (Aladdin, 99.99%), strontium acetate (Aladdin, 99%), titanium butoxide (Aladdin, 99%), magnesium acetate tetrahydrate (Aladdin, 99%), manganese acetate tetrahydrate (Aladdin, 99%), zirconium propoxide solution (Aladdin, 70%) were dissolved in an acetate (Aladdin, 99.5%) solvent and a 2-Methoxyethanol solvent. Ammonia and acetylacetone were used as stabilizers and the concentration of the precursor solution was adjusted to 0.2 M by the addition of 2-Methoxyethanol, followed by continuous stirring for six hours at room temperature for complete dissolution. The solution was filtered by syringe filter with a pore size of 0.2 μm to obtain clear and transparent precursor solutions. The clarified and stabilized precursor solution was spin-coated on a Pt substrate. The rotational speed was 4500 rad/min for 30 s. The obtained films were pyrolyzed at 200 °C, 300 °C, and 450 °C for five minutes to remove the residual organic matter. Finally, a rapid heat treatment was carried out at 640 °C for two minutes to obtain a film with a thickness of about 160 nm and a uniform and dense surface (Supplementary Fig. 6).

## Characterizations

An X-ray diffractometer (Smartlab, Rigaku) with Cu Kα radiation was used to characterize the crystal structure of these prepared films. A field emission scanning electron microscope (FE-SEM; JSM 7610 F Plus) was used to characterize the film thickness and surface microstructure. Characterization of crystal interiors used HR-TEM (Thermo Fisher Talos F200X). A commercial scanning probe microscope (MFP-3D, Asylum Research) was applied for PFM measurements. The dielectric properties of thin films were measured by the impendence analyzer (Agilent 4294, USA). Dielectric constant and loss tangent are measured in the frequency range from 1 kHz to 1 MHz. The ferroelectric properties were collected using a ferroelectric workstation (Precision Premier II, Radiant Technologies Inc., USA) at 1 kHz and room temperature.

## Reporting summary

Further information on research design is available in the Nature Portfolio Reporting Summary linked to this article.

# Data availability

All data used are available within this paper and Supplementary Information. Further information can be acquired from the corresponding authors upon reasonable request.

# Code availability

All codes in this work are available from the corresponding author upon reasonable request.

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

## Acknowledgements

This work was supported by Basic Science Center Program of NSFC (Grant No. 52388201, C.-W.N.), the NSF of China (Grant No. 52372121 and 52002300, Z.-H.S.), the National Key Research and Development Program of China (No. 2023YFB3812200, H.-X.L.), the Major Research Plan of NSFC (Grant No. 92066103, Z.-H.S.).

## Author contributions

These authors contributed equally: W.L. and Z.-H.S. Z.-H.S. and C.-W.N conceived and designed the research; W.L. prepared the ceramic films and wrote the first draft of the paper; R.-L.L. established the machine learning model; X.-X.C. performed the phse-field simulations; M.-F.G. performed PFM measurements; J.-M.G. conducted TEM testing; Z.-H.S. and C.-W.N. reviewed and edited the paper with contributions from H.H., Y.S., H.-X.L., L.-Q.C.

## Competing interests

The authors declare no competing interests.
