## [Peer Review File · Nature Communications]

Generative learning facilitated discovery of high-entropy ceramic dielectrics for capacitive energy storageREVIEWER COMMENTS

Reviewer #1 (Remarks to the Author):

This manuscript proposes a generative machine learning approach to accelerate the discovery and development of high-entropy dielectric ceramics. The high-entropy strategy of inducing local disorder enables comprehensive regulation of the energy storage properties in high-performance dielectric ceramic materials. Based on the phase-field simulations and a relatively limited database, the application of generative machine learning significantly reduces the trial-and-error space in experiments, shortening the experimental cycle time. The approach is validated through directed experiments, establishing a theoretical-to-experimental methodological path. The paper is well-structured, and the narrative is ingenious and innovative. With minor corrections/revisions to address the issues outlined below, this manuscript could be recommended for publication in Nature Communications.

1. The background information is somewhat insufficient, considering the existence of polymer-based dielectric materials in addition to ceramic-based ones. It is suggested to include one or two sentences to differentiate between these two classes of dielectric materials. Critical reviews such as Chem. Rev. 2021, 122, 3820, Chem. Soc. Rev. 2021, 50, 6369, and Nat. Nanotechnol. 2024, doi.org/10.1038/s41560-023-01416-3, can be considered. Additionally, it is recommended to emphasize ceramic dielectrics in the title.
2. Why choose BMT as the base system and those specific elements for doping?
3. How were the 77 initial data points selected, and why exactly 77?
4. In machine learning, large-scale training data are typically required to ensure result reliability. The authors are requested to provide more details and clearly state the advantages of their study with small-scale training data to demonstrate the effectiveness of their approach more clearly.
5. In forward speculation, machine learning integration models are employed to predict the energy storage performance of materials. Please provide more detailed information on this regression model.
6. Please explain the role of Bayesian optimization in the overall integrated model and describe in detail how the hyper-parameters are determined through Bayesian optimization.
7. In this machine learning-guided design process, how did the authors consider the crystal structure of these high-entropy ceramic materials? Additionally, it would be helpful to know whether the deep learning model in this paper is applicable to other systems or if it can be extended to polymer dielectrics with semi-crystalline or amorphous natures.

Reviewer #2 (Remarks to the Author):

The authors initially employ phase field simulation to illustrate that high entropy design is an effective method for optimizing the balance among maximum polarization, remanent polarization, and breakdown strength. Subsequently, they develop a generative learning-based framework to expedite the discovery of high-entropy ceramic materials with high energy density. The optimized compositions discovered are validated through directed experiments, demonstrating an energy density as high as 156 J cm⁻³. Overall, this paper presents intriguing new exploration methods to pursue high-energy dielectrics. I recommend its publication, contingent upon addressing the following concerns:

1. In line 26, "ultrafast" should be corrected to "ultrahigh."

2. According to the phase field simulation (Fig. 1d), theoretically, the larger the S_{config} , the higher the energy density. Is there a limit of the entropy that one can practically realize in a material? Could introducing more dopant elements lead to higher entropy and energy density?
3. Line 143, "high-energy design" should be changed to "high entropy design"
4. The scale bar in Supplementary Fig. 6 is not sufficiently clear.
5. The authors attribute the decreased permittivity, increased breakdown strength, and decreased remanent polarization to grain refinement and the increase in the proportion of the amorphous phase. However, no direct evidence has been provided.
7. C2 sample exhibits the lowest leakage current (Supplementary Fig. 7), contradicting its lowest efficiency in Fig 4b.
8. The evidence from PFM is insufficient to draw the conclusion of greatly reduced amplitude. Plotting the amplitude as a function of applied voltage to show changes in the "butterfly-shaped curves would clarify this.
9. In line 307, the PE loop of BMT appears acceptable. However, the PE loop of other samples C1-5 seems anomalous. As the field approaches 0 from negative values, the polarization reaches a plateau.
10. High entropy materials is a hot topic with many interesting systems being discovered recently. It is suggested to compare with the counterparts in the literature and show the position of the current reported materials.

Reviewer #3 (Remarks to the Author):

The paper "Generative learning facilitated discovery of high-entropy dielectrics for capacitive energy storage" is an original work based on generative algorithms applied to high-entropy dielectrics.

This is an interesting article that tries to design high-entropy dielectrics using state-of-the-art machine learning methods, with the experimental data. The whole story is complete and the results are good. In principle, this is a sound work. However, the reviewer found the framework of machine learning is very similar to a recently published work, Rao Z, et al. Machine learning-enabled high-entropy alloy discovery. *Science* 378, 78-85 (2022). The reviewer does not think it is a problem to borrow something from other work and apply it to different materials. However, the reviewer found that the authors seem just copy the framework but ignore many details. Regards to different materials or different datasets, active learning can not be the same, although the authors did not mention the word "active learning" in this paper.

The reviewer thinks the paper can be published after answering the following questions:

1. The author claims "Here, based on phase-field simulations and limited experimental data, we propose a generative learning approach to accelerate the discovery of high-entropy dielectrics in a practically infinite exploration space of over 10^{14} combinations." Why it is over 10^{14} combinations?
2. The authors claim "Through only 5 sets of targeted experiments", if it is true, the results will be very surprisingly good. But why does it work so well? Compared to many other works, it always takes several iterations to get the best results (if you have an interest, you can

check the paper of Prof. T. Lookman in the last 10 years). The authors provided many analyses of the materials, which is very good. However, analysis or discussion about the ML framework is much less. The authors must explain why it works so well in the case (i.e. much fewer experiments than previous papers on active learning)

3. Also about question 2, if the authors can answer the previous question, it will give some insight to the whole community to know how to optimize active learning in different cases. This will also improve the quality of this paper. So I think question 2 is very important. The authors must convince the reviewer why it works so well in this case. The materials scientists do not like 'black box'.

4. For the generative model, the authors show a decrease of error during the training, this is good but not direct. Can the authors show an example (like the first 10 rows) of the reconstruction compared with the original input? Then it will be very clear to see the difference of the reconstruction and the input.

5. Why do you use a classification model instead of a regression model in the latent space?

6. Since you add a classification mode and also regression models for the prediction, why do you claim the design is based on a generative model? The regression and classification models are not generative models.

7. Why do you use GMM and MCMC? There is no explanation in the paper.

8. Why do you use ANN and lgbt? There is also no explanation about this? Can you use more basic methods like Gaussian regression models?

9. What is ranking policy, I did not see any explanation about this in the paper.

10. Fig 2b is very confusing. What is the meaning of the purple circles? Is the colors of the generative alloys connected with the color bar? I did not see any purple region in the color bar.

11. How to explain Fig 2c, why the upright regions have very high uncertainty?

12. The authors claim 'we have made extensive efforts to find high-entropy systems with suitable elements and their contents by trial-and-error methods.' This is not very convincing. As materials scientists, we have domain knowledge, it is not pure trial and error, right?

13. Supplementary Fig. 2 is very strange, the error increases with the training, how to explain this?

14. How about the training results of ANN and lgbt, the reviewer does not see them.

15. In the methods part, what is MMD loss? Why do the authors use MMD loss here? Sounds strange.

16. Also in the methods part, the authors claim 'which is the weighted sum of maximum mean discrepancy between the z and prior distribution and binary cross-entropy of input-output pair', however, the reviewer does not see 'sum' in equation 8. Are the authors sure that equation 8 is right? Please carefully check the ML methods part.

17. If it is possible, the reviewer would like to see the original code of the machine learning part.

Point-by-point Response Letter

Dear Referees:

We sincerely appreciate the reviewers' time and efforts for carefully reviewing our manuscript, providing valuable comments and suggestions. We have answered and analyzed the experiments mentioned in the article as well as the machine learning aspects of the manuscript. In what follows, we have made point-to-point responses (marked in **blue**) to all reviewers' comments. The changes in the revised manuscript are marked in **red**, and the original text in *grey*. We hope our efforts address all the concerns, and the current version satisfies the criteria of *Nature Communications*.

Reviewer #1 (Remarks to the Author):

This manuscript proposes a generative machine learning approach to accelerate the discovery and development of high-entropy dielectric ceramics. The high-entropy strategy of inducing local disorder enables comprehensive regulation of the energy storage properties in high-performance dielectric ceramic materials. Based on the phase-field simulations and a relatively limited database, the application of generative machine learning significantly reduces the trial-and-error space in experiments, shortening the experimental cycle time. The approach is validated through directed experiments, establishing a theoretical-to-experimental methodological path. The paper is well-structured, and the narrative is ingenious and innovative. With minor corrections/revisions to address the issues outlined below, this manuscript could be recommended for publication in *Nature Communications*.

Response:

We would like to thank the reviewer for the positive comments on our manuscript. We have carefully addressed the reviewer's concerns, and our point-by-point replies to the comments are listed below.

1. The background information is somewhat insufficient, considering the existence of

polymer-based dielectric materials in addition to ceramic-based ones. It is suggested to include one or two sentences to differentiate between these two classes of dielectric materials. Critical reviews such as Chem. Rev. 2021, 122, 3820, Chem. Soc. Rev. 2021, 50, 6369, and Nat. Nanotechnol. 2024, doi.org/10.1038/s41560-023-01416-3, can be considered. Additionally, it is recommended to emphasize ceramic dielectrics in the title.

Response:

We thank the reviewer for the valuable comments, which certainly help us to improve the quality of our manuscript. We have made changes in the sections of Introduction and Title in the revised manuscript.

[Revision to manuscript] (Page 1):

*Generative learning facilitated discovery of high-entropy **ceramic** dielectrics for capacitive energy storage*

[Revision to manuscript] (Page 3):

*Dielectric capacitors capable of storing and releasing charges by electric polar dipoles are the essential elements in modern electronic and electrical applications such as hybrid electric vehicles, portable electronic devices as well as power pulse systems, owing to much higher power density than the electrochemical counterparts^{1, 2}. **However, both ceramics possessing high dielectric constant and polymers featured by high breakdown strength face the dilemma that the energy density U_e is much lower than that of chemical energy storage devices such as batteries^{3, 4}. The lower energy density of dielectric materials greatly limits their applications and developments towards miniaturization and integration in the new era of the Internet of Things⁵. Therefore, it is of great significance to develop dielectric capacitors with higher energy density.***

2. Why choose BMT as the base system and those specific elements for doping?

Response:

Thanks for the reviewer's careful consideration of the option of material systems in our manuscript. When choosing the material system and doping elements, we

consider the following three aspects of the ferroelectricity of BMT, the advantages of BMT system, and the possible role of doping elements.

(1) $\text{Bi}(\text{Mg}_{1/2}\text{Ti}_{1/2})\text{O}_3$ (BMT) has been used as an end-element in wide-temperature ceramic capacitors in recent years, and has been used in the exploration of energy storage materials because of its ferroelectricity (*Applied Surface Science*, 285, 744-747, 2013; *Ceramics International*, 40, 5327-5332, 2014; *Journal of Materials Science*, 59, 2757-2775, 2024). A large number of studies have shown that BMT has ultrahigh polarization intensity comparable to BiFeO_3 , and better voltage resistance and insulation than BiFeO_3 (*Journal of Materials Chemistry C*, 7, 13632-13639, 2019; *Ceramics International*, 49, 37238-37244, 2023; *ACS Applied Materials and Interfaces*, 16, 3654-3664, 2024). As we described on the Page 5 of the manuscript, we choose BMT because of its strong ferroelectric features and relatively good stability as pristine matrix to design high-entropy dielectrics by simultaneous multi-element doping of its A-site and B-site. Thus, the purpose of selecting BMT for this work is to make full use of its high polarization and further improve the breakdown strength through high entropy design, so as to obtain higher energy density.

(2) For doping elements, we selected and designed those specific elements based on the role of doping elements in the existing literatures and our previous works. For examples, we found that Mn can inhibit the valence state change of Ti^{4+} and reduce the leakage current path formed by free electron hopping between Ti^{4+} and Ti^{3+} . By reducing the leakage coefficient, it makes the carrier potential well deeper, thus increasing the breakdown strength (*Small*, 18, 2106209, 2022). The doping of La reduces the grain size of the ceramics and makes the ceramics dense during sintering thereby increasing the resistivity of the high-entropy ceramics. Meanwhile, with the increase of La concentration, the decrease of grain size produced more grain boundaries, which increases the ceramic resistance and activation energy. And La doping leads to stronger relaxation behavior and disordered structure, which is conducive to the improvement of breakdown strength and energy storage properties of high-entropy ceramics (*Journal of the American Ceramic Society*, 106, 6641-6653, 2023). Higher Sr mole fraction leads to a decrease in grain size, and the P - E loop at room temperature

evolves from a hysteresis loop to a linear thin loop with increasing Sr content, indicating a transition from the ferroelectric phase to the paraelectric phase (*Journal of Materials Chemistry C*, 10, 3876-3885, 2022; *Materials Research Express*, 6, 026310, 2018). Zr ions can inhibit the electron hopping generated by Ti ions in different valence states, thus improving the breakdown strength. In addition, Zr⁴⁺ has stronger chemical stability than Ti⁴⁺, and Zr substitution can induce relaxation behavior when the Zr concentration is less than 10 % mol (*Ceramics International*, 47, 32357-32363, 2021). As discussed above, we briefly summarize the possible effects of elements in the following Table R1.

Table R1. The possible effects of elemental doping

dopant elements	Possible effects
Mn	Enhances breakdown strength and promotes PNR formation
La	Enhances breakdown strength and promotes PNR formation
Sr	Enhances breakdown strength and facilitates ferroelectric to paraelectric transition
Zr	Enhanced breakdown strength, induced relaxation behavior
Ca	Reduces crystallinity and increases breakdown strength
Hf	Reducing residual polarization and inducing a relaxing ferroelectric phase

All of the above indicate that those doping elements have the potential to enhance the energy density by affecting one or more key performance parameters, such as the breakdown strength and polarization. However, their role in BMT and how they interact with each other are not clear, which are something we want to address in this work by machine learning. Therefore, we used BMT as the pristine system and doped it with the corresponding elements for our experiments to find the optimal combination of high-entropy dielectrics.

3. How were the 77 initial data points selected, and why exactly 77?

Response:

We thank the reviewer's professional question on the initial data. The data selection of initial experimental points in this work is based on the randomness and the distribution of elements, and the stability of the machine learning model.

Firstly, these 77 data sets are random variations that we have designed based on the gradual increase in the type and number of elements, with some compositions designed on the base of experience and the results of existing experiments that have been carried out. Secondly, these variations cover every element as much as possible because of the large number of element types and the possibility of changes in elemental content. For example, we carried out the experiments of individual doping of BMT system by Zr, Mn, Sr, and La respectively, and then we carried out the mixed doping of various elements such as La, Mn, Zr, and Sr.

In addition, we also used 77 initial data to explore whether these data can meet the encoder-decoder model to reconstruct the components. As illustrated in Supplementary Fig. 1, the gradual convergence of the smooth curve indicates that the model can effectively handle a large amount of high-entropy component information after being trained on these 77 sets of data. In this work, we found that exactly 77 sets of data make the model perform well. It should be noted that for different research systems and objectives, the demand for data volume is different.

Supplementary Fig. 1 | Reconstruction loss of the encoder-decoder structure. The

components of the A and B sites are encoded respectively, with the same structure. The loss function is a weighted sum of cross entropy and maximum mean discrepancy, which levels off as the number of training epoch increases.

4. In machine learning, large-scale training data are typically required to ensure result reliability. The authors are requested to provide more details and clearly state the advantages of their study with small-scale training data to demonstrate the effectiveness of their approach more clearly.

Response:

We thank the reviewer for the professional questions about machine learning datasets. Yes, in machine learning, large-scale training data can ensure the reliability and generalization ability of models. However, for high-entropy dielectrics and even other material systems, acquiring large-scale training data may face challenges such as high experimental costs and lengthy time requirements. Therefore, finding a suitable machine learning algorithm with significant application and development potential in the realm of material small data is crucial (*npj Computational Materials*, 9, 42, 2023). As an unsupervised or partially supervised machine learning framework, generative learning can fit the posterior distribution of target samples, thereby improving the quality of generated samples. In this work, we leverage generative model to generate more new data by learning the underlying patterns or distributions from 77 initial data. Then, 2144 groups of component data with high energy storage potential are sampled from the prior probability of the classifier through the autoencoder and GMM-MCMC algorithm. This approach enables rapid generation of new material designs using minimal data by learning from existing data and descriptors of dielectric physical properties. Thus, as our results show, the generative model helps us solve the small sample problem well and is successfully used to design high-entropy ceramic dielectrics.

5. In forward speculation, machine learning integration models are employed to predict the energy storage performance of materials. Please provide more detailed information on this regression model.

Response:

We thank the reviewer for the valuable comments, and more details about the regression model are disclosed below.

This study utilizes regression models, including ANN and LightGBM, to investigate the correlation between the composition and energy storage performance of high-entropy dielectrics. Additionally, we semi-quantitatively predict candidate high-entropy dielectrics based on this relationship. To improve the model's generalization ability and avoid overfitting due to insufficient data, ensemble learning integrates the prediction results of 50 models to estimate prediction uncertainty. The hyperparameters of these models were obtained and adjusted using Bayesian optimization (BO). The data was divided into training and validation sets with a ratio of 0.85:0.15, and the same random seeds remained unchanged. Using the best performing model as an example, we utilized a neural network with n layers of the same fully connected layer for prediction in the ANN model. The specific hyperparameters, detailed in the table R2, were determined using BO. The optimizer selected Adam, the weight attenuation was set to 0.0001, the number of iterations was set to 3000, and we used the Kaiming initialization method to ensure network stability. The neural network training processes are implemented in the PyTorch framework, and the loss function is Mean Absolute Percentage Error (MAPE).

Table R2. The hyperparameters of the BO results for ANN

learning_rate	number of hidden layer neurons	n	batch_size
0.001	206	10	54

For LightGBM, it serves as a framework that implements the Gradient Boosting Decision Tree (GBDT) algorithm, offering efficient parallel training capabilities for predicting energy density. Similarly, all 12 hyperparameters within LightGBM undergo optimization using the BO to minimize the MAPE as the error function (Table R3).

Table R3. The hyperparameters of the BO results for LightGBM

learning_rate	colsample_bytree	num_leaves	n_estimators	...
0.1	1	12	395	...

As shown in Fig. R1 (a) and (b), it can be seen that after BO, the accuracy of the prediction model of all the dataset has reached more than 88% in entire dataset, indicating that the energy density of the dielectrics can be positively predicted based on the composition and physical descriptors of high-entropy dielectrics. It is important to note that the limited amount of data and the high complexity of high-entropy systems can cause the model to overlook the important features, resulting in underfitting or overfitting of model performance. Therefore, in the forward inference process, we rank the prediction results and introduce model uncertainty to identify high-performance combinations.

Fig. R1 The fitting results of experimental energy density and predicted energy density of two models: (a) ANN and (b) Light GBM.

[Revision to manuscript] (Page 19):

ANN, inspired by neural networks, excels at learning complex patterns, while LightGBM is a tree-based gradient boosting framework, efficient for handling high-dimensional features. This work integrates both to enhance predictive accuracy and robustness in regression tasks, among which the main hyperparameters were optimized by random searches followed by 10 rounds of Bayesian Optimization (BO). To eliminate

the dimensional impact and enhance the stability of the regression model, all atomic properties were normalized. Moreover, predicted values and uncertainty were calculated by top fifty models from BO to reduce the incidental errors. The program was written using Pytorch and Sklearn.

6. Please explain the role of Bayesian optimization in the overall integrated model and describe in detail how the hyper-parameters are determined through Bayesian optimization.

Response:

We thank the reviewer for the comments. BO is a method for optimizing black-box functions by building a probabilistic model for hyperparameter optimization (*Information Science for Materials Discovery and Design*, 45-75, 2016; *Advanced Theory and Simulations*, 2, 1900110, 2019). It builds a prior model based on the objective function and updates the model when new data is observed. By continuously utilizing prior information and observation data, we can gradually converge to the objective function reaching the optimal solution or its approximate solution, thereby selecting the most promising parameters for evaluation. The BO in this article takes the negative MAPE loss function as the optimization goal, optimizing 4 hyperparameters as shown in Table R2 in ANN, and optimizing 12 hyperparameters as shown in Table R3 in LightGBM. The following Table R4 shows performance comparisons of integrated models using BO and not using BO. At the same time, we have added information about BO in the Methods section.

Table R4. Performance comparison of integrated models using BO and not using BO

	R^2 : Using BO	R^2 : Unused BO
ANN	0.734	0.905
LightGBM	0.487	0.885

[Revision to manuscript] (Page 19):

The program was written using Pytorch and Sklearn. BO is used in the bayes_opt

library in python.

7. In this machine learning-guided design process, how did the authors consider the crystal structure of these high-entropy ceramic materials? Additionally, it would be helpful to know whether the deep learning model in this paper is applicable to other systems or if it can be extended to polymer dielectrics with semi-crystalline or amorphous natures.

Response:

We thank the reviewer's professional questions on the machine learning model and its potential applications. We will answer those questions separately.

(1) For crystal structure considerations, the types and contents of elements in high-entropy ceramics have a large variation room. Therefore, during the machine learning design process, we introduced a large number of physically-based crystal structure-related descriptors to improve the accuracy of the predictions (e.g., Supplementary Table 5), such as the tolerance factor, the ionic radius of each element, the atomic volume of the AB site, the electronegativity of the AB site, etc. (*Acta Materialia*, 209, 116815, 2021).

(2) For the potential applications, we believe that deep learning models can be also applied to other material systems. For polymer dielectrics, we can try but may need to embed different physical knowledge in specific domains, e.g., (Structural unit, average degree of polymerization, relative molecular weight, etc.). Techniques such as descriptor construction, symbolic regression, rule establishment, and graph neural networks for handling graph data have been proven to enhance model robustness. For example, Wu *et al.* used machine learning trained on limited polymer property data and used trained molecular design algorithms to identify quantitative structure-property relationships and discover new polymers with high thermal conductivity (*npj Computational Materials*, 5, 66, 2019). Kim *et al.* employ a nature-mimicking optimization method, the genetic algorithm, in tandem with ML-based predictive models to design polymers that meet practically useful, but extreme, property criteria

(i.e., glass transition temperature, $T_g > 500$ K and bandgap, $E_g > 6$ eV) (*Computational Materials Science*, 186, 110067, 2021).

Therefore, we believe that our deep learning model has great potential in accelerating the design of other systems including semi-crystalline polymers by introducing different descriptors and domain knowledge.

Supplementary Table 5 Physical descriptors in the machine learning model.

Descriptor	Explanation
$R_A(\text{\AA})$	Ionic radii of A-site (12-coordination)
$R_B(\text{\AA})$	Ionic radii of B-site (12-coordination)
$AR_A(\text{\AA})$	Atomic radius of A-site element
$AR_B(\text{\AA})$	Atomic radius of B-site element
$A-O_A(\text{\AA})$	Ideal A–O bond distance
$A-O_B(\text{\AA})$	Ideal B–O bond distance
V_A	Atomic volume of A-site element
V_B	Atomic volume of B-site element
VWR_A	Crystallographic van der Waals radii of A-site element
VWR_B	Crystallographic van der Waals radii of B-site element
PE_A	Period of A-site element in element period table
PE_B	Period of B-site element in element period table
W_A	Relative atomic mass of A-site element
W_B	Relative atomic mass of B-site element
AN_A	Atomic number of A-site element in element period table
AN_B	Atomic number of B-site element in element period table
EN_A	A-site electronegativity (Pearson 1988)
EN_B	B-site electronegativity
EI_A	First energy ionization of A-site element
EI_B	First energy ionization of B-site element
EA_A	Electron affinity of A-site element
EA_B	Electron affinity of B-site element
t	Tolerance factor calculated by ionic radii
μ	Octahedral factor calculated by ionic radii
E	Entropy calculated by compositions

Reviewer #2 (Remarks to the Author)

The authors initially employ phase field simulation to illustrate that high entropy design is an effective method for optimizing the balance among maximum polarization, remanent polarization, and breakdown strength. Subsequently, they develop a generative learning-based framework to expedite the discovery of high-entropy ceramic materials with high energy density. The optimized compositions discovered are validated through directed experiments, demonstrating an energy density as high as 156 J cm⁻³. Overall, this paper presents intriguing new exploration methods to pursue high-energy dielectrics. I recommend its publication, contingent upon addressing the following concerns:

Response:

We are thankful to the reviewer for the appreciation of our work and valuable comments. We have carefully addressed the reviewer's concerns with extra data and discussion. The modifications of the text, figures and tables have been highlighted. Please see the detailed content below.

1. In line 26, "ultrafast" should be corrected to "ultrahigh."

Response:

We really appreciate the reviewer for careful reading and suggestion of the manuscript. We have carefully considered the inappropriate formulation and have revised it in the manuscript.

[Revision to manuscript] (Page 2):

*Dielectric capacitors offer great potential for advanced electronics due to their **ultrahigh** power densities, but their energy density still needs to be further improved.*

2. According to the phase field simulation (Fig. 1d), theoretically, the larger the S_{config} , the higher the energy density. Is there a limit of the entropy that one can practically realize in a material? Could introducing more dopant elements lead to higher entropy

and energy density?

Response:

We thank the reviewer's insightful comment. Yes, both simulation and experimental results show that the conformational entropy is closely related to energy density. However, entropy values can reach critical values in real situations, and it is not the case that higher entropy values are better for the energy density.

Theoretically, the entropy value varies with the number and content of elements. This can be seen according to the conformational entropy formula:

$S = R \ln N$, where R , N (M) and x_i (x_j) denote

the ideal gas constant, the atomic species at the positive/anionic position and the elemental molar concentration, respectively. In fact, the entropy value increases with the increase of elemental species and content, and the configurational entropy reaches the maximum value when the molar concentrations of all elements are equal. (*Journal of Applied Physics*, 133, 110904, 2023; *InfoMat*, 5, 12, 2023). However, in practice, as the number of elements increases, the preparation of high-entropy materials with high quality will become more difficult. For example, for sol-gel method in this work, the stable preparation of precursors and subsequent heat treatment will face great challenges. Therefore, in the actual experiments, it is difficult to increase the entropy infinitely. More importantly, pursuing entropy too high does not necessarily lead to better energy storage performance, which will be discussed as follows.

According to the existing research, there seems to be an inflection point between the entropy value and the energy density. For example, Wang *et al.* mentioned that the energy density reaches a maximum of 8.8 J/cm³ when the entropy value is 2.1R, and as the entropy value continues to increase, the energy density starts to decrease (*Journal of Materials Chemistry A*, 11, 4937-4945, 2023). Qiao *et al.* reported that the energy density reaches 7.16 J/cm³ when the doping content is 0.2, and as the doping content increases, the energy density also starts to decrease (*Chemical Engineering Journal*, 477, 147167, 2023). Yang *et al.* mentioned in his article that the maximum energy density 182 J/cm³ is reached at an entropy value of 1.6R. As the entropy value continues

to increase up to 1.72R, the energy density decreases (*Nature Materials*, 21, 1074-1080, 2022). The phase field simulation of Yang *et al.* shows that there is an inflection point in the energy density as the entropy value increases, and subsequent experiments also show that the maximum energy density of 178.1 J/cm³ is reached at an entropy value of 1.2R. As the entropy value continues to increase, the energy density starts to decrease (*Nature Energy*, 8, 956-964, 2023). There, when entropy increases to a certain value, it may have a negative effect to reduce the energy density. It should be noted that the optimal entropy value is highly dependent on the material system and preparation technology.

It is well known that the energy density of a dielectric is determined by the breakdown strength (E_b), the maximum polarization (P_m) and the residual polarization (P_r). However, for most dielectrics, the polarization tends to be negatively correlated with the breakdown strength (*IEEE Transactions on Electron Devices*, 50, 1771-1778, 2003). With the increase of entropy value, E_b is likely to increase and the polarization P will increase. When E_b and $P_m - P_r$ reach a certain state of trade-off, the energy density reaches the maximum. If the entropy value continues to increase, there will be a rapid decrease in $P_m - P_r$, which will lead to the decreasing of the energy density (*Nature Energy*, 8, 956-964, 2023).

In summary, theoretical entropy value could be adjusted by changing the elemental species and content. Reaching a certain degree of high entropy will be conducive to the improvement of energy storage density. However, in practice, the entropy value cannot increase without limit, and there is an inflection point between the energy density and the entropy value in experiments.

3. Line 143, "high-energy design" should be changed to "high entropy design"

Response:

We really appreciate the reviewer for careful reading and suggestion of the manuscript. We have carefully reconsidered the improper statement and have revised it in the manuscript.

[Revision to manuscript] (Page 7):

The improvement of energy storage performance by *high entropy design* concluded by our simulations results is also consistent with the existing experiments^{12, 24, 31}.

4. The scale bar in Supplementary Fig. 6 is not sufficiently clear.

Response:

We thank the reviewer for this valuable suggestion on our images. Supplementary Fig. 6 has been updated in the revised manuscript.

[Revision to Supplementary Information] (Page 8):

Supplementary Fig. 6 | SEM images of the films of (a) BMT and (b)-(f) C-n (n=1,2,3,4,5).

5. The authors attribute the decreased permittivity, increased breakdown strength, and decreased remanent polarization to grain refinement and the increase in the proportion of the amorphous phase. However, no direct evidence has been provided.

Response:

We thank the reviewer for the questions about the experiment. In order to explain the changes of dielectric performance more comprehensively, we added the characterization of TEM and analyzed those variations from the following three aspects of SEM, GIXRD and TEM.

Firstly, according to the SEM image (Supplementary Fig. 6), we can see that the grain size of C-n (n=1,2,3,4,5) decreases compared with BMT. And we use nano measurement software to count the grain size, as shown in Supplementary Fig. 7. It can be seen more intuitively that the average grain size of pure BMT is 181 nm, while the grain size of C-n (n=1,2,3,4,5) is significantly smaller, and the average grain size is about 30 ~ 60 nm.

Secondly, according to the GIXRD diagram (Fig. 3a), we can see that the intensity of the (110) peaks of C-n films are obviously weaker than that of the BMT, and the diffraction peaks are gradually broadened, which indicates that the crystal size in the sample is gradually refined. At the same time, the gradual broadening of the peaks can be regarded as the infinite fineness of the grains, and ultimately between a crystalline phase and the amorphous phase of the excessive situation that is the quasi-crystalline state (*Applied Physics Letters*, 77, 2587-2589, 2000; *Journal of Materials Chemistry C*, 7, 13632-13639, 2019; *Journal of Materials Chemistry A*, 7, 17797-17805, 2019). We also fitted the integration of the main peaks of XRD to calculate the total area of the peaks and the area of the amorphous peaks. Crystallinity was obtained using the reference strength ratio (RIR) method, as shown in Fig. R2. It can be seen that the degree of crystallinity of BMT is about 87.5%, while the crystallinity of composition C-3 is the lowest, which is about 50% (*Analytical Methods*, 9, 2415-2424, 2017; *Minerals*, 10, 13, 2020). From this, it can be judged that the proportion of amorphous phase of C-n films are increased.

Thirdly, we analyzed the BMT and C-3 samples by transmission electron microscopy (Thermo Fisher Talos F200X), shooting at a voltage of 200 kV. The lattice diffraction fringes can be observed in the area of the yellow circles in Supplementary Fig. 8a and 8b, which are thus determined to be crystalline, and the rest amorphous. The HR-TEM results could show the percentage of local crystalline and amorphous phases of BMT and C-3. As shown in Supplementary Fig. 8c, with the increase of entropy from BMT to C-3, the percentage of amorphous phase increased from 10% to 45%.

Therefore, the grain refinement and the increase in the proportion of the

amorphous phase in high-entropy films are evidenced by the results of SEM, GIXRD and TEM. Small grain size with more grain boundaries and more amorphous phase are beneficial to the decreased permittivity, increased breakdown strength, and decreased remanent polarization (*Journal of the American Ceramic Society*, 101, 5487-5496, 2018; *Physical Chemistry Chemical Physics*, 21, 16207-16212, 2019). Supplementary Fig. 7 and Supplementary Fig. 8 in the Supplementary Information have been updated as shown below. In the meantime, we have made changes to page 12 of the manuscript.

Supplementary Fig. 6 | SEM images of the films of (a) BMT and (b)-(f) C-n ($n=1,2,3,4,5$).

Fig. 3a GI-XRD patterns of BMT and C-n ($n=1,2,3,4,5$) films.

Fig. R2 Calculated crystallinity of each composition based on the reference strength ratio (RIR) method.

[Revision to manuscript] (Page 12):

This is also evident from the scanning electron microscope (SEM) of different films after annealing (Supplementary Fig. 6), which show denser microstructures with nanograins down to a few nanometers in Supplementary Fig. 7. As the high resolution transmission electron microscope (HR-TEM) results shown in Supplementary Fig. 8a and 8b, the lattice diffraction fringes can be observed in the area of the yellow circles, which are thus determined to be crystalline, and the rest amorphous. With the increase of entropy from BMT to C-3, the percentage of local amorphous phase increased from 10 percent to 45 percent (Supplementary Fig. 8c). More details about the experimental preparation and characterization are described in Methods.

[Revision to Supplementary Information] (Page 8 and 9)

Supplementary Fig. 7 | Grain size distribution of the films of (a) BMT and (b)-(f) C-n ($n=1,2,3,4,5$).

Supplementary Fig. 8 | HR-TEM images of lattice fringes (dotted yellow area) with different directions in (a) BMT and (b) C-3. (c) Fractions of the local crystalline and amorphous-like phases in the samples of BMT and C-3.

6. C-2 sample exhibits the lowest leakage current (Supplementary Fig. 7), contradicting its lowest efficiency in Fig 4b.

Response:

We thank the reviewer for this important question and apologize for not displaying them clearly in the manuscript, which we will explain in more detail below:

The leakage current data shown in Supplementary Fig. 9 only applies to the case below 1000 kV/cm. If it is higher than 1000 kV/cm, it cannot be detected by our experimental apparatus. Thus, only the leakage current data below 1000 kV/cm are displayed. Whereas C-2 in our original Fig. 4b shows efficiency only above 1000 kV/cm, it can be seen that the efficiency of C-2 is slightly lower than the other four compositions. We have now added the data for the C-2 samples below 1000 kV/cm to Fig. 4b, which have all been updated in the revised manuscript.

At the same time, we display the efficiency of BMT and C-n (n=1, 2, 3, 4, 5) under the electric field of 1000 kV/cm in Fig. R3. It can be seen that the efficiency of C-1 is slightly lower than the other compositions under the small electric field, and C-2 is slightly higher than the other four compositions.

Fig. R3 Efficiency of BMT and C-n (n=1,2,3,4,5) at the electric fields below 1000 kV/cm.

[Revision to manuscript] (Page 28):

Fig. 4b Energy density and efficiency at the electric fields up to E_b .

7. The evidence from PFM is insufficient to draw the conclusion of greatly reduced amplitude. Plotting the amplitude as a function of applied voltage to show changes in the "butterfly-shaped curves would clarify this.

Response:

We are very grateful to the reviewer for this professional suggestion. We have added the butterfly-shaped curves to demonstrate the greatly reduction in amplitude.

We measured the amplitude and phase of a ± 1 V triangular wave (frequency 350 kHz) by leaving the tip at a fixed position on the sample. As shown in Supplementary Fig. 11, characteristic square hysteresis loops in phase and butterfly-shaped loops in amplitude are plotted to represent the changes in polarization and piezoelectric amplitude, respectively. It can be seen that the coercive electric field and piezoelectric amplitude of C-3 sample are lower than that of BMT sample. We have also made changes in the revised manuscript and Supplementary Fig. 11 has been updated in the supplementary information.

[Revision to manuscript] (Page 13):

To understand the huge difference of polarization response between BMT and C-n films, we also characterized the local domain structures by piezo-response force microscopy

(PFM) in two representative films of BMT and C-3 sample, *as shown in Supplementary Fig. 10 and Supplementary Fig. 11*. Compared to the pure BMT, greatly reduced domain size and amplitude can be clearly identified in high-entropy C-3 film, which can be an important reason for the slim P-E loops.

[Revision to Supplementary Information] (Page 12):

Supplementary Fig. 11 | PFM phase (top panel) and amplitude (bottom panel of (a,c) BMT thin film and (b,d) C-3 thin film.

8. In line 307, the PE loop of BMT appears acceptable. However, the PE loop of other samples C1-5 seems anomalous. As the field approaches 0 from negative values, the polarization reaches a plateau.

Response:

We thank the reviewer for this nice question. Yes, the plateaus in P-E loops of C1-5 samples are different from the regular ferroelectric hysteresis loop of BMT sample. After analyzing experimental data and consulting a lot of literatures, we found this plateau mainly appears in the P-E loop under high electric field (> 2000 kV/mm), which

is why no platform appears in BMT. Thus, we think this abnormal phenomenon may be due to multi-element doping causing leakage current of the film and other reasons.

Ferroelectricity can be recognized directly by the hysteresis return line. The polarization reaches a plateau as the field converges from negative to zero, indicating that our polarization value tends to a stable one. This may be because as the applied electric field decreases, the induced electric field generated inside the sample due to leakage conduction and thus the external electric field cannot cause a flip of domains inside the sample, which results in a plateau in the polarization value region. Some studies reported that this may be due to the presence of various multivalent cations such as Mn ions, Ti ions, heterovalent substituents and oxygen vacancies, which leads to an increase in leakage current. (*Journal of the American Ceramic Society*, 97, 27, 2014; *Applied Physics Letters*, 93, 032902, 2008).

In addition, the BMT system in this work is similar to BiFeO₃ in that both have extremely high polarization. As the *P-E* loop of BiFeO₃ system shown in Fig. R4, the loop cannot be closed and there is also a gap due to the polarization back-switching and point defects, and hence leakage conductance of the film (*Materials Research Bulletin*, 147, 111617, 2022). Thus, similar to BiFeO₃, with increasing electric field, the plateau of the *P-E* loop will eventually tend to be completely horizontal, which may be attributed to the presence of valence altering cations within the system, oxygen vacancies, which leads to an increase in the leakage current (*Journal of the American Ceramic Society*, 97, 27, 2014). Thus, we think that the main reason for the plateau of our *P-E* loop may be due to the leakage current, which is related to the choice of our system, the selection of elements and the preparation process.

Fig. R4 P - E loops of BiFeO_3 ceramics sintered at $760\text{ }^\circ\text{C}$ before (non-poled) and after poling at 80 kV/cm of DC electric field (poled) (*Journal of Applied physics*, 112, 064114, 2012).

9. High entropy materials is a hot topic with many interesting systems being discovered recently. It is suggested to compare with the counterparts in the literature and show the position of the current reported materials.

Response:

We thank the reviewer for this constructive suggestion, which is invaluable to the integrity of our article.

According to the reviewer's comment, we have summarized some experimental results as shown in Supplementary Fig. 13. It shows the relationship between the energy density and breakdown strength of different dielectric materials based on some common systems. Some ceramic bulks are indicated in the red circle at the lower left corner of the figure, and some ceramic films are indicated in the red shaded area. High-entropy systems are represented by hollow data points and non-high-entropy systems are displayed by solid data points. It can be seen that our work is in a relatively leading position among the currently available reports on both high-entropy and non-high-entropy ceramic bulks and films. We have revised and added to the Supplementary Information accordingly.

[Revision to Supplementary Information] (Page 14):

Supplementary Fig. 13 | Comparison of energy storage properties of C-3 high-entropy ceramic film with reported experimental results, where hollow centres are high-entropy data points and solid centres are non-high-entropy data points.

Reviewer #3 (Remarks to the Author):

The paper “Generative learning facilitated discovery of high-entropy dielectrics for capacitive energy storage” is an original work based on generative algorithms applied to high-entropy dielectrics.

This is an interesting article that tries to design high-entropy dielectrics using state-of-the-art machine learning methods, with the experimental data. The whole story is complete and the results are good. In principle, this is a sound work. However, the reviewer found the framework of machine learning is very similar to a recently published work, Rao Z, et al. Machine learning-enabled high-entropy alloy discovery. *Science* 378, 78-85 (2022). The reviewer does not think it is a problem to borrow something from other work and apply it to different materials. However, the reviewer found that the authors seem just copy the framework but ignore many details. Regards to different materials or different datasets, active learning can not be the same, although the authors did not mention the word “active learning” in this paper.

The reviewer thinks the paper can be published after answering the following questions:

Response:

Thank you for reviewing our submitted manuscript carefully and providing the valuable comments. In the field of materials science, the development of new high-entropy materials faces a vast space for exploration due to the explosive growth of material components. Designing reliable machine learning methods to efficiently design specific materials has become an important research direction. Yes, Rao Z. *et al.* successfully designed high-entropy alloys using generative and ensemble models, providing an effective method for the component design of highly complex high-entropy materials. This work is very enlightening and provides us with a new idea for the design of dielectric materials. Therefore, we learned from this good method with a similar framework. At the same time, based on the characteristics of dielectric materials design and experimental preparation process, we also improved the machine learning model in terms of data processing, model selection, descriptor design, and uncertainty.

In the following, we will address the reviewer’s concerns with extra data and discussion and spare no effort to ensure the quality and accuracy of the manuscript to meet your expectations.

1. The author claims “Here, based on phase-field simulations and limited experimental data, we propose a generative learning approach to accelerate the discovery of high-entropy dielectrics in a practically infinite exploration space of over 10^{14} combinations.” Why it is over 10^{14} combinations?

Response:

Thanks for the important questions, and we apologize that we did not explain them clearly in the manuscript. Our study is based on the ABO_3 perovskite structure, where the A-site and B-site can accommodate different elements. In this work, the A-site can accommodate four different elements, while the B-site is set to be replaced by five possible elements. The compositions of these elements must satisfy the condition that the total content of elements in both A-site and B-site is equal to 1. To systematically explore the possible combinations of elements, we used an integer programming approach and carefully calculated the composition space. We assumed that the content of each component is retained to the second decimal place. Table R5 below lists the possible number of elements at each location and their corresponding combinations. Based on this calculation, the size of the possibilities we obtained is on the order of 10^{11} . We have corrected and briefly described it in the manuscript.

Table R5. Number of possible combinations of different elements number

Number of elements	Number of possible combinations
2	101
3	5151
4	176851
5	4598126
This work	$4598126 * 176851 = 813183181226$

[Revision to manuscript] (Page 5):

Taking 77 sets of experimental results as initial data, we build a generative learning

model based on an encoding-decoding architecture with data reconstruction and artificial neural network (ANN) to find the potentially optimal high-entropy combinations^{28, 29}. *The existing small sample data is then augmented with probabilistic sampling, where the elemental content of the A- and B-positions are retained to two decimal places and each position is summed up to equal 1. Thus, a possible space of nearly 10^{11} combinations is constructed to search for optimal combinations that satisfy the high entropy criterion. Then, we screen out top five compositions of prediction results among more than 2,000 candidates, and five groups of targeted experiments are conducted to verify their potential in energy storage performance.*

2. The authors claim “Through only 5 sets of targeted experiments”, if it is true, the results will be very surprisingly good. But why does it work so well? Compared to many other works, it always takes several iterations to get the best results (if you have an interest, you can check the paper of Prof. T. Lookman in the last 10 years). The authors provided many analyses of the materials, which is very good. However, analysis or discussion about the ML framework is much less. The authors must explain why it works so well in the case (i.e. much fewer experiments than previous papers on active learning).

Response:

We are grateful to the reviewer for this valuable question. We have carefully read Professor T. Lookman’s papers on materials design using active learning (such as, *Nature Communication*, 7, 11241, 2016; *Acta Materialia*, 170, 109-117, 2019; *npj Computational Materials*, 5, 21, 2019; *Advanced materials*, 30, 1702884, 2018; *Computational Materials Science*, 129, 311-322, 2017). There is no doubt that the active learning model is an efficient materials design method, which improves the accuracy of prediction results by continuously iterating experiments. Active learning is usually a trade-off between exploration and exploitation. It expands the data set by feeding back the most information-rich points to the data set, thereby efficiently improving the generalization ability of the model. This method has been widely used

in ferroelectric, piezoelectric, alloy and other material systems. Typical active learning methods are based on simple machine learning agent models that directly learn the relationship between data and performance. For the examples of alloys and ceramics, the experimental process is relative stable and the performance is less affected by the experimental conditions, making it easier to conduct iterative experiments. Therefore, for ideal active learning with stable preparation process, the more experimental iterations, the better the performance is likely to be.

However, in this work, we use sol-gel method to prepare high-entropy films, where the precursor solution becomes unstable with the increase of elements. Therefore, the data quality from the iterative experiments is difficult to guarantee. If the results of the directed experiments differ greatly from the theoretical fact due to the problems of the precursor preparation, it would bring the negative feedback to the active learning model and thus the search direction of the optimal solution may be misleading. We think that the poor performance of many active learning models is probably due to the problem of bad experimental feedback. Because we cannot avoid the problem of the instability of precursor solutions in a short time, we hope to achieve the optimal design of high-entropy dielectrics with several isolated directed experiments in one batch. To achieve better model prediction without iteration, we improved the machine learning model by designing a complex generative model, considering a number of physical descriptors and designing a different rank policy based on the idea of active learning. We believe the good performance of our model stems from the following points:

(1) We designed an encoder respectively for the A-site and B-site of the high-entropy dielectrics to better capture the information of the complex components. Based on this two-encoder strategy, 2144 sets of data were sampled from the high energy density area through the GMM-MCMC algorithm to achieve the generation of candidate materials. As can be illustrated in Supplementary Table 3 and 4, we reconstructed the data with good reliability. And our framework of machine learning models such as reconstructed error maps, energy storage density classification models, and Gaussian density distributions all show the feasibility of this process, as shown in Supplementary Figs. 1, 2 and 3. It should be mentioned that the randomness and

uniformity of doping elements are fully considered in the design and selection of the initial data, which is also helpful to the final prediction results.

(2) We introduced some ferroelectric information descriptors of dielectrics, such as structure, composition and entropy. These descriptors enable a more complete description of the dielectric properties, improving the model's training performance, as shown in Supplementary Table 5. Meanwhile, the accuracy of the prediction model of all the dataset has reached more than 88% in entire dataset, indicating that the energy density of the dielectrics can be positively predicted based on the composition and physical descriptors of high-entropy dielectrics (Fig. R5).

Fig. R5 The fitting results of experimental energy density and predicted energy density of two models: (a) ANN and (b) Light GBM.

(3) Different from active learning, we adopted an uncertainty-based ranking strategy to guide the material design process based on the BO-based integrated model. This strategy utilizes a machine learning ensemble model to predict energy density, with model uncertainty based on the variance of the ensemble model predictions. Usually in active learning, samples with high predicted energy density and high uncertainty are selected for experimental verification to improve model generalization ability. Different from the active learning, we choose samples with high accuracy and low uncertainty by sorting and weighting. The specific sorting strategy is answered in details in question 9. The purpose is to obtain component points with small uncertainty and high energy density at once. This improves the efficiency of experimental decision-making and the

credibility of results, and alleviates problems caused by unstable experimental processes.

Therefore, to avoid the complexity and instability of the experimental process that may result in negative feedback for active learning and relieve the difficulty of small data in the field, we conducted a one-time parallel experiment to complete the verification based on generative learning-based strategy. This framework includes a generative model based on encoder-decoder, embedded ferroelectric physical descriptors, integrated regression models, ranking strategies, and uncertainty-based stability assessment. The algorithmic framework searches for clusters with high energy density and generates new candidate dielectrics with similar distributions. It then converts inaccuracies in material data predictions into qualitative rankings. Finally, we successfully performed five parallel experiments to verify the effectiveness of this method and obtain the high-energy dielectric with high energy density. However, we have to mention that despite the above improvements to the machine learning model, we still cannot guarantee the complete reliability of the results, which is why there are still samples with poor energy storage performance in the five directed experiments. In addition, poor energy storage performance of our samples such as C-4 film may also be the problem of sample preparation. All in all, we use active learning-based model to achieve relatively efficient and accurate design on the base of small sample data to avoid preparation process influences as much as possible.

[Revision to manuscript] (Page 10):

...to guide the discovery of desirable compositions. In traditional active learning strategies, the uncertainty tends to favor the space of compositional combinations whose predicted values have higher variance. We have improved uncertainty trade-off with the goal of predicting dense, stable compositional data in one go. We rank combinations by λ , as this strategy magnifies the gap between different candidates. The ranking-based strategy ensures that candidate portfolio selection is less affected by model inaccuracies and provides a systematic way to combine model predictions and uncertainty.

3. Also about question 2, if the authors can answer the previous question, it will give some insight to the whole community to know how to optimize active learning in different cases. This will also improve the quality of this paper. So I think question 2 is very important. The authors must convince the reviewer why it works so well in this case. The materials scientists do not like 'black box'.

Response:

Thank you for the valuable suggestions provided by the reviewer. Your viewpoints are crucial as they determine the direction for intelligent material design in the entire field of material science. Explainability in machine learning typically involves understanding of deep model interpretations, the impact of training data, and insights from domain knowledge (*ACM Computing Surveys*, 55, 1-33, 2023; *Nature Reviews Materials*, 8, 241-260, 2023; *Nature Reviews Physics*, 3, 422-440, 2021). In conjunction with our framework, we will elaborate on this and provide some of our own insights on how machine learning can be applied to experimental material design.

(1) The importance of active learning and experimental iteration is self-evident. Theoretically, active learning centres on the trade-off between predictive value and uncertainty. It extends the dataset by feeding back to the most informative points of the dataset, which effectively improves the generalization of the model. Through continuous online learning, the performance of the model will become better and better. Therefore, more positive feedback from iterative experiment results will ensure that the model performs better and better. Additionally, advancements in reinforcement learning and the development of large-scale models enable the possibility of automated laboratories, which is also a future direction (*Available at SSRN*, 4707535; *Nature Communications*, 14, 1403, 2023).

(2) The significance of uncertainty in materials cannot be ignored. To ensure repeatability and robustness in experiments, multiple experiments are necessary for each material system to avoid random errors. Moreover, factors such as processes and environmental conditions can introduce errors in experimental results. How to use machine learning algorithms to identify noise and errors in experimental data is also

very important. The bias in small material datasets may also lead to poor generalization of machine learning models. Therefore, when designing machine learning models, it is essential to consider the characteristics of the data and embed domain knowledge to enhance model robustness. Techniques such as descriptor construction, symbolic regression, rule establishment, and graph neural networks for handling graph data have been proven to enhance model robustness. Additionally, well-designed strategies such as ranking policies can mitigate biases resulting from small datasets.

(3) In model design, integrating advanced deep learning techniques to develop feasible material design schemes, such as encoder-decoder architectures, generative adversarial networks (GANs) (*Computational Materials Science*, 220, 112064, 2023), and diffusion models (*Nature Machine Intelligence*, 5, 1466-1475, 2023), is paramount. These generative models can effectively address the scarcity of material data. Furthermore, transfer learning (*Computer Methods in Applied Mechanics and Engineering*, 405, 115852, 2023) and reinforcement learning can also be employed for this purpose. Specifically, we can design optimal solutions based on material requirements.

In summary, based on the difficulties in the preparation of high-entropy sol-gel process and the adjustment of the uncertainty of the active learning model, we have improved the whole machine learning process as we discussed in the last question. Firstly, we propose a unique dual-encoder generative model strategy for the inverse generation of materials using well-designed initial data, and apply the dielectric-specific descriptors and unique ranking strategy to the regression model to obtain a targeted experimental scheme. Finally, we experimentally verify the correctness of the design scheme using five parallel experiments in one batch.

4. For the generative model, the authors show a decrease of error during the training, this is good but not direct. Can the authors show an example (like the first 10 rows) of the reconstruction compared with the original input? Then it will be very clear to see the difference of the reconstruction and the input.

Response:

We thank the reviewer for the valuable suggestions. We show an example of the first 10 rows in the Supplementary Table 3 and Supplementary Table 4 below. It is noted that the reconstructed data is essentially consistent with the input original data when preserved to two decimal places.

[Revision to manuscript] (Page 9):

*Using the elemental composition information of every initial film as the input data, a latent feature space z that can be used to potentially represent the dielectric information variables is generated, and then decoding z can regenerate the reconstructed composition. **Supplementary Fig. 1, Supplementary Table 3 and Supplementary Table 4 show the reconstructed results of the compositions.** The model is also analyzed for its ability to extract high-entropy compositions represented as low-dimensional latent variables, as shown in Supplementary Fig. 1, where the smooth curve indicates that the model eventually stabilizes.*

[Revision to Supplementary Information] (Page 18):

Supplementary Table 3 Original data of the first 10 rows.

Bi	La	Sr	Ca	Ti	Mg	Mn	Zr	Hf
0.90	0.10	0.0	0.0	0.45	0.45	0.05	0.05	0.0
0.90	0.10	0.0	0.0	0.40	0.40	0.10	0.10	0.0
0.90	0.10	0.0	0.0	0.35	0.35	0.15	0.15	0.0
0.90	0.10	0.0	0.0	0.30	0.30	0.20	0.20	0.0
0.90	0.10	0.0	0.0	0.25	0.25	0.25	0.25	0.0
0.90	0.05	0.05	0.0	0.40	0.40	0.10	0.10	0.0
0.94	0.03	0.03	0.0	0.40	0.40	0.10	0.10	0.0
0.94	0.03	0.03	0.0	0.35	0.35	0.15	0.15	0.0
0.9	0.10	0.00	0.0	0.70	0.10	0.00	0.10	0.1
0.95	0.00	0.05	0.0	0.50	0.50	0.00	0.00	0.0

Supplementary Table 4 Reconstruction data of the first 10 rows.

Bi	La	Sr	Ca	Ti	Mg	Mn	Zr	Hf
0.8980	0.1020	0.0000	0.0000	0.4532	0.4472	0.0458	0.0538	0.0000
0.8980	0.1020	0.0000	0.0000	0.4011	0.3964	0.1008	0.1017	0.0000
0.8980	0.1020	0.0000	0.0000	0.3509	0.3447	0.1548	0.1497	0.0000
0.8980	0.1020	0.0000	0.0000	0.2978	0.2941	0.2079	0.2002	0.0001
0.8980	0.1020	0.0000	0.0000	0.2471	0.2449	0.2583	0.2493	0.0004
0.8986	0.0476	0.0537	0.0000	0.4011	0.3964	0.1008	0.1017	0.0000
0.9404	0.0270	0.0325	0.0001	0.4011	0.3964	0.1008	0.1017	0.0000
0.9404	0.0270	0.0325	0.0001	0.3509	0.3447	0.1548	0.1497	0.0000
0.8980	0.1020	0.0000	0.0000	0.6907	0.0964	0.0000	0.1074	0.1054
0.9524	0.0003	0.0473	0.0000	0.4971	0.5020	0.0007	0.0000	0.0002

5. Why do you use a classification model instead of a regression model in the latent space?

Response:

We appreciate the reviewer's valuable comment. Firstly, high-entropy ceramic dielectrics have a complex composition-structure-property mapping, which makes comprehensive exploration challenging. Therefore, achieving accurate quantitative composition-structure-property relationships by regression models is often difficult ($R^2 < 0.2$). On the other hand, classification models are better suited for handling such complex nonlinear problems, demonstrating outstanding training performance and higher interpretability ($R^2 > 0.74$). In the latent space, we try to map material compositions directly to energy storage performance. Our objective is to analyze whether a certain component belongs to the high-energy density group (such as $> 65 \text{ J cm}^{-3}$ in this work). Then, we could conduct subsequent MCMC sampling based on this classifier, performing probability sampling around high energy density regions, generating 2144 sets of data.

6. Since you add a classification mode and also regression models for the prediction, why do you claim the design is based on a generative model? The regression and

classification models are not generative models.

Response:

We thank for the reviewer's valuable questions and apologize for not explaining them clearly in our manuscript. Yes, the machine learning framework in this work includes not only generative models, but also regression and classification. However, as we mentioned in the article, the generative model solves the most central challenge of small-data problem, which is also the basis for subsequent classification and regression. Therefore, we believe that generative learning is the key to this work, so we use generative learning in the title to emphasize its importance contribution in this machine learning strategy.

A generative model is a statistical model that learns the process of generating data and can produce new data samples, as opposed to simply classifying or regressing existing data. It is important to note that regression and classification models are typically classified as discriminative models rather than generative models. This work develops a machine learning structure that combines generative and discriminative models. Specifically, a variational autoencoder model is used to generate new data samples similar to high-entropy dielectrics with high energy density. The approach involves encoding-decoding and classifier-based MCMC sampling, which is an innovative approach described in the manuscript. The original data is utilized to train a regression model. The generated data is then predicted and ranked to attain the candidate materials with high energy density.

To more clearly describe the differences and roles of three models in this work, we have revised the manuscript correspondingly and emphasized the description of the generative model.

[Revision to manuscript] (Page 8):

As displayed in Fig. 2a, the machine learning framework consists of three parts: (i) the generative model with generation of the latent space z (ii) classification and sampling of compositions, and the predictive model of (iii) forward inference and inverse design.

7. Why do you use GMM and MCMC? There is no explanation in the paper.

Response:

We thank the reviewer for the comment. GMM and MCMC are two commonly used statistical modeling methods that play an important role in dealing with complex data distribution. By using GMM to model the probability density distribution of the data, MCMC can perform sampling and thus achieve effective sampling of high energy density materials for high-entropy materials. In GMM, it is assumed that the data is a mixture of multiple Gaussian distributions, with each distribution being a component. The mixture of these components is adjusted by weights to obtain the probability distribution of density in the latent space. This study utilizes GMM to derive the data probability prior density distribution of the high energy density group in the classifier. Additionally, MCMC, a sampling method based on Markov chain Monte Carlo, is employed to generate samples by constructing a smoothly distributed Markov chain with the same density distribution as the aforementioned.

To explain different roles of GMM and MCMC, we have also made changes in the methods section of the manuscript.

[Revision to manuscript] (Page 19):

The GMM assumes the data is composed of multiple Gaussian distributions, which was used to model and estimate the distribution density of z . The optimal number of Gaussian clusters is usually determined by the empirical elbow method. ANN classifier was trained with two layers of simple neural network to identify raw samples with a high energy density. In order to make the classifier more precious, k -fold cross-validation was used. Then, we utilized Metropolis-Hasting MCMC in the latent space z based on Markov chain, aiming to generate numerous data according to accept-reject sampling.

8. Why do you use ANN and lgbt? There is also no explanation about this? Can you use more basic methods like Gaussian regression models?

Response:

We thank the reviewer for the important comments and suggestions. As shown in Fig. R6(a) and (b), both ANN and LightGBM (*Advances in neural information processing systems*, 30, 2017) demonstrate an accuracy exceeding 87% in all dataset. Meanwhile, they possess numerous hyperparameters, and Bayesian optimization allows for obtaining multiple robust machine learning models. In fact, we also used Gaussian process regression (GPR) and found it to be less effective as shown in Fig. R6(c) below. Moreover, its number of hyperparameters is relatively small, which is not conducive to Bayesian optimization.

Fig. R6 The fitting results of experimental energy density and predicted energy density of two models: (a) ANN and (b) Light GBM (c) GPR.

9. What is ranking policy, I did not see any explanation about this in the paper.

Response:

We thank the reviewer for the comment. We apologize for not explaining this clearly in the manuscript. In order to reduce dependence on the absolute value of energy

density, focus more on the relative size between prediction results, and reduce the impact of noise in the data on model performance, we therefore considered changing the absolute value into a ranking size. Note that in order to obtain a broader component space and consider the problem of model overfitting, we sort the uncertainties from small to large to reduce the sensitivity to noise and improve the model's qualitative prediction ability to unknown data. Then we use λ to weight the ranking of predicted values and the ranking of uncertainty, as follows:

where α denotes predicted value, β denotes uncertainty. Uncertainty represents the variance of the predicted values of the integrated model and can provide information about the level of confidence in the predicted results, helping us to better understand the predictive significance of the model.

[Revision to manuscript] (Page 19 and 20):

To eliminate the dimensional impact and enhance the stability of the regression model, all atomic properties were normalized. Moreover, predicted values and uncertainty were calculated by top fifty models from BO to reduce the incidental errors. We then measured the value of λ through a ranking strategy, as shown in Equation 9, where α denotes predicted value, β denotes uncertainty. The program was written using Pytorch and Sklearn. BO is used in the bayes_opt library in python.

(9)

10. Fig 2b is very confusing. What is the meaning of the purple circles? Is the colors of the generative alloys connected with the color bar? I did not see any purple region in the color bar.

Response:

We thank the reviewer for the comment. We have re-explained Fig. 2b in more details below.

The GM-generated potential space z is visualized by principal component analysis

(PCA) downscaling, as shown in Fig. 2b, where the red dots denote the original experimental data, corresponding to the color bar on the right side. Purple circles denote the 2144 candidate sets of high-energy-density potential data sampled from classifier ($U_e > 65 \text{ J/cm}^3$). We have also revised and supplemented it in the manuscript accordingly.

Fig. 2b Latent space distribution of the different compositions, where the color of the data points in the latent space indicates their predicted value of energy density.

[Revision to manuscript] (Page 10):

Based on 77 sets of BMT-based experiment results as initial data, we generate 2,144 sets of high-performance systems with energy densities greater than 65 J cm^{-3} , and then select the top five sets for targeted experiments. *The GM-generated potential space z is visualized by principal component analysis (PCA) downscaling, as shown in Fig. 2b, where the red dots denote the original experimental data, corresponding to the color bar on the right side. Purple circles denote the 2144 candidate sets of high-energy-density potential data sampled from the classifier ($U_e > 65 \text{ J/cm}^3$). Pentagrams with different colors indicate the new five components generated by model predictions (C- n , $n=1,2,3,4,5$). Their elemental species and contents, entropy values and uncertainties are shown in Supplementary Table 6. The five new compositions are located in the middle dense region (inside the yellow circle), indicating that the energy density of the compositions in this region is more in line with our expectations. The*

relationship between the entropy value and the normalized U_e of each composition in the candidate space predicted by the regression model is shown in Fig. 2c.

11. How to explain Fig 2c, why the upright regions have very high uncertainty?

Response:

We thank the reviewer for the questions. Firstly, the uncertainty represents the variance of the prediction value of the integrated model and the confidence level of the prediction result. As the components become more and more complex and the entropy value increases, the dielectric materials have more degrees of freedom and possible combinations. The prediction ability of the energy density naturally decreases and the uncertainty increases. According to the Phase Field section of the manuscript, a positive correlation between entropy value and energy density is also revealed.

12. The authors claim ‘we have made extensive efforts to find high-entropy systems with suitable elements and their contents by trial-and-error methods.’ This is not very convincing. As materials scientists, we have domain knowledge, it is not pure trial and error, right?

Response:

We thank the reviewer for this professional question. Yes, we rely on combining professional experience to design initial experiments. For example, it has been reported that BMT has a stable ionic configuration with a high polarization value (*Materials*, 9, 935, 2016), the elements Sr and Mn have the effect of increasing the breakdown strength of the system (*Small*, 18, 2106209, 2021; *Journal of Materials Chemistry C*, 10, 3876-3885, 2022), and the elements La and Zr have the effect of decreasing the residual polarization, etc (*Journal of the American Ceramic Society*, 106, 6641-6653, 2023; *Ceramics International*, 47, 32357-32363, 2021). Therefore, we have developed a series of experiments based on an empirically orientated and trial-and-error approach. Thus, we have made changes to unreasonable expressions in the revised manuscript.

[Revision to manuscript] (Page 8):

To experimentally realize the high-entropy design in dielectrics, based on our knowledge and experience in the field of materials, we carry out an expert-orientated trial-and-error method to find high-entropy systems with suitable elements and their contents. Taking $\text{Bi}(\text{Mg}_{0.5}\text{Ti}_{0.5})\text{O}_3$ as initial matrix, we have prepared 77 systems of $\text{Bi}_{(1-a-b-c)}\text{La}_a\text{Sr}_b\text{Ca}_c(\text{Mg}_{0.5}\text{Ti}_{0.5})_{1-d-e-f}\text{Mn}_d\text{Zr}_e\text{Hf}_f\text{O}_3$, including 48 sets of high-entropy combinations by introducing different elements in the A-site and B-site, as detailed at Supplementary Table 2.

13. Supplementary Fig. 2 is very strange, the error increases with the training, how to explain this?

Response:

Thanks for this question and we apologize that we did not explain Supplementary Fig. 2 clearly. In fact, in Supplementary Fig. 2, the observed phenomenon may be attributed to the results of employing five-fold cross-validation (*Computational Materials Science*, 171, 109203, 2020). In five-fold cross-validation, the dataset is divided into five parts, with four parts sequentially chosen as the training set and the remaining part as the validation set for each iteration. As the partition of the training and validation sets is random, it may lead to differences in the characteristics of certain training sets compared to others, thereby influencing the performance evaluation of the model.

It should be noted that we used R^2 , the coefficient of determination, to measure the model's explanation of the data variance, rather than error. In some cases, R^2 may increase during the model training process as the model better fits the variance of the training data. However, this does not imply an increase in the model's prediction error, as our concern lies in the overall performance of the model on the entire dataset.

Supplementary Fig. 2 | The results of five-fold cross-validation using a simple ANN classifier. This classifier is divided by an energy density threshold of 65 J/cm^3 . The batch size is 16 and the learning rate is 5×10^{-3} .

14. How about the training results of ANN and lgbt, the reviewer does not see them.

Response:

We thank the reviewer for this comment. The training results of ANN and lgbt are shown in Fig R7 a and b. It can be seen that after Bayesian optimization, the accuracy of the prediction model of all the dataset has reached more than 88%, indicating that the energy density of the dielectrics can be positively predicted based on the composition and structural information of high-entropy dielectrics.

Fig. R7 The fitting results of experimental energy density and predicted energy density of two models: (a) ANN and (b) Light GBM.

15. In the methods part, what is MMD loss? Why do the authors use MMD loss here? Sounds strange.

Response:

We thank the reviewer for the comment. To clarify, Maximum Mean Discrepancy (MMD) is a measure of the difference between two probability distributions (*Advances in Neural Information Processing Systems*, 32, 2019). The MMD is defined by a feature map $\varphi: S \rightarrow H$, H is defined as the reproducing kernel Hilbert space.

where P and Q are two distributions of different dataset. Such loss function is widely used in generative models, especially in variational autoencoders, to measure the difference between the distribution of samples in the latent space and a given prior distribution. In the encoder-decoder structure of this article, a method is needed to ensure that the encoded sample distribution matches the given prior distribution. By using MMD loss, VAE can be used to minimize the sample distribution in the latent space with the given prior distribution. The difference between the prior distributions is used for training. It can also be seen from Supplementary Table 3 and 4 that the material composition is reconstructed based on MMD loss and cross-entropy loss.

16. Also in the methods part, the authors claim ‘which is the weighted sum of maximum mean discrepancy between the z and prior distribution and binary cross-entropy of input-output pair’, however, the reviewer does not see ‘sum’ in equation 8. Are the authors sure that equation 8 is right? Please carefully check the ML methods part.

Response:

We are very grateful to the reviewer for this important question and we have carefully checked equation in the methodology. In fact, the binary cross entropy calculation formula is as follows (*Proceedings of the 23rd international conference on Machine learning*, 161-168, 2006):

As shown in the above formula, it carries a “-” of its own, so the formula in the weighted sum should be fine, and we apologize to the reviewer for the trouble we caused by not presenting it clearly.

17. If it is possible, the reviewer would like to see the original code of the machine learning part.

Response:

We thank the reviewer for the comment. We open source the code to <https://github.com/lowwo/high-entropy-dielectrics-design>.

REVIEWER COMMENTS

Reviewer #1 (Remarks to the Author):

The authors have addressed the reviewer's feedback, significantly improving the quality and clarity of the manuscript. I recommend accepting it without change.

Reviewer #2 (Remarks to the Author):

The authors have fully addressed the reviewers' concerns. No further revision is needed and the paper can be published as it is.

Reviewer #3 (Remarks to the Author):

The reviewer still has the following comments, and the paper can be accepted after addressing them:

1. In Line 90, 'large errors' is not very accurate, 'High bias' is a better word.
2. In Line 188, 'Markov Chain-Montecarlo Sampling' should be 'Markov Chain-Monte Carlo Sampling'
3. Why do you use PCA in the generative model? You can also get the latent space without the PCA. Is there any difference?
4. In Fig.2(b), the Pentagrams are difficult to identify, especially the red one. Can you make these Pentagrams more clear?

Reviewer #1 (Remarks to the Author):

The authors have addressed the reviewer's feedback, significantly improving the quality and clarity of the manuscript. I recommend accepting it without change.

Response:

We are deeply grateful to the reviewer for reviewing our manuscript and agreeing to publish it.

Reviewer #2 (Remarks to the Author):

The authors have fully addressed the reviewers' concerns. No further revision is needed and the paper can be published as it is.

Response:

We are very thankful to the reviewer for the approval and recommendation of our manuscript.

Reviewer #3 (Remarks to the Author):

The reviewer still has the following comments, and the paper can be accepted after addressing them:

Response:

We are very appreciative of the reviewer' suggestions and recommendations for our manuscript. We will provide reasonable explanations to the reviewer' questions point by point.

1. In Line 90, “large error” is not very accurate, “High bias”is a better word.

Response:

We really appreciate the reviewer for careful reading and suggestion of the manuscript. We have carefully considered the inappropriate formulation and have revised it in the manuscript.

[Revision to manuscript] (Page 4):

*When data is scarce, machine learning models would encounter problems such as poor generalization ability, overfitting and **high bias**¹⁹⁻²¹. Generative learning, different...*

2. In Line 188, “Markov Chain-Montecarlo Sampling” should be “Markov Chain-Monte Carlo Sampling”

Response:

We really appreciate the reviewer for careful reading and suggestion of the manuscript. We have carefully reconsidered the improper statement and have revised it in the manuscript.

[Revision to manuscript] (Page 9):

*Because the space of C-n compositions is too large with about 1011 possible combinations, we analyze known high-performance compositions in the low-dimensional space by Gaussian mixture model (GMM) (Supplementary Fig. 3) and **Markov Chain-Monte Carlo Sampling** (MCMC) to inversely generate unseen other compositions with similar high performance^{35, 36}.*

3. Why do you use PCA in the generative model? You can also get the latent space without the PCA. Is there any difference?

Response:

We thank the reviewer for the valuable comments. We apologize for not explaining clearly in the manuscript the role of PCA in our overall machine learning framework. Principal Component Analysis (PCA) is an important statistical method for dimensionality reduction and plays an important role in data visualisation as well as knowledge extraction (*Nature Biotechnology*, 26, 303-304, 2008). In this work, we reconstruct the A-bit and B-bit components using two encoder-decoder architectures, generating two latent spaces z , so the total dimensionality is four. In order to make the complex data easier to understand and interpret, and to discover hidden patterns, we further visualised it using PCA downscaling to two dimensions. The results of the

downscaled visualisation of the latent space are shown in Fig. 2b, where we can see that the five component points screened are distributed in a more concentrated area, suggesting that there is a greater likelihood of excavating high energy storage ceramics in this region.

Therefore, PCA in our machine learning framework is not used for latent space generation, but as a way to downscale the data in the latent space for visualisation purposes.

4. In Fig. 2(b), the Pentagrams are difficult to identify, especially the red one. Can you make these Pentagrams more clear?

Response:

We are very grateful to the reviewer for the suggestions. We have refined Fig. 2b and revised some expressions in the manuscript.

[Revision to manuscript] (Page 10):

*The GM-generated potential space z is visualized by principal component analysis (PCA) downscaling, as shown in Fig. 2b, where the **blue squares** denote the original experimental data, corresponding to the color bar on the right side. Purple circles denote the 2144 candidate sets of high-energy-density potential data sampled from the classifier ($U_e > 65 \text{ J/cm}^3$). **Solid spheres** with different colors indicate the new five components generated by model predictions (C- n , $n=1,2,3,4,5$).*

[Revision to manuscript] (Page 26):

Fig. 2b Latent space distribution of the different components, purple circles represent the generated 2144 sets of high-performance data, blue squares represent the original experimental data, and solid spheres of different colors represent the five new sets of components predicted by the model

REVIEWERS' COMMENTS

Reviewer #3 (Remarks to the Author):

I have no more comments. I recommend the publication.

Reviewer #3 (Remarks to the Author):

I have no more comments. I recommend the publication.

Response:

We are deeply grateful to the reviewer for reviewing our manuscript and agreeing to publish it.